# Patchy and widespread distribution of bacterial translation arrest peptides associated with the protein localization machinery

Keigo Fujiwara [1,2] ✉, Naoko Tsuji[1,2], Mayu Yoshida[1,2], Hiraku Takada[1,2] & Shinobu Chiba [1,2] ✉

Regulatory arrest peptides interact with specific residues on bacterial ribosomes and arrest their own translation. Here, we analyse over 30,000 bacterial genome sequences to identify additional Sec/YidC-related arrest peptides, followed by in vivo and in vitro analyses. We find that Sec/YidC-related arrest peptides show patchy, but widespread, phylogenetic distribution throughout the bacterial domain. Several of the identified peptides contain distinct conserved sequences near the C-termini, but are still able to efficiently stall bacterial ribosomes in vitro and in vivo. In addition, we identify many arrest peptides that share an R-A-P-P-like sequence, suggesting that this sequence might serve as a common evolutionary seed to overcome ribosomal structural differences across species.

Regulatory nascent chains exert their cellular functions while they are still nascent polypeptides[1–3]. They induce programmed ribosomal stalling by interacting with ribosomal residues located near the peptidyl transferase center (PTC), the mid-tunnel region within the nascent polypeptide exit tunnel (NPET), and, occasionally, on the ribosomal surface[4–8]. Thus, they are also called ribosome arrest peptides or simply arrest peptides[1]. Translation arrest generally occurs under a specific intracellular condition, thus allowing arrest peptides to respond to changes in the intracellular environment to serve as sensors of the feedback gene regulation.

A class of bacterial arrest peptides, such as SecM, MifM, and VemP, is involved in the feedback regulation of genes encoding components of the protein localization machinery[9–11]. *Escherichia coli* SecM and *Vibrio alginolyticus* VemP monitor the Sec protein secretion pathway, in which SecA and SecDF facilitate protein translocation in ATP- and proton-motive force-depending manners, respectively[12–15]. *Bacillus subtilis* MifM monitors the YidC membrane protein insertion pathway, in which YidC serves as an "insertase" for a class of membrane proteins[16–20].

The *secM* gene encodes a protein with the N-terminal Sec-dependent signal sequence[21] and the C-terminal arrest sequence ($F_{150}$xxxxWIxxxxGIRAGP$_{166}$; x indicates residues whose identities are unimportant for the arrest)[22], and is co-transcribed with its downstream *secA* gene (Fig. 1a)[23]. A stem-loop structure sequesters the Shine–Dalgarno (SD) sequence of *secA*[24]. The stalled ribosome on *secM* interferes with the stem-loop structure, thus allowing SecA synthesis[22,25]. Engagement of the SecM nascent chain with the active Sec translocation machinery leads to the arrest cancellation[26]. Thus, a malfunction of the Sec machinery results in a prolonged arrest of SecM, leading to *secA* induction[25]. MifM and VemP feedback-regulate the downstream *yidC2* (*yqjG*) and *secD2/F2* genes, respectively, in a similar fashion[10,11]. As SecM, MifM, and VemP are substrates of the protein-translocation pathway that they monitor, they are also called "monitoring substrates"[27].

The lack of sequence similarity among the arrest sequences has hampered the identification of novel arrest peptides based on conventional approaches. To overcome this obstacle, we recently established an in silico screening system to find open reading frames (ORFs) that possibly encode monitoring substrates based on features shared by known monitoring substrates[28]. Our previous search across 449 bacterial genomes identified three arrest peptides, i.e., ApcA and ApdA

[1]Faculty of Life Sciences, Kyoto Sangyo University, Motoyama, Kamigamo, Kita-Ku, Kyoto 603-8555, Japan. [2]Institute for Protein Dynamics, Kyoto Sangyo University, Kyoto, Japan. ✉e-mail: kigfujiwara@cc.kyoto-su.ac.jp; schiba@cc.kyoto-su.ac.jp

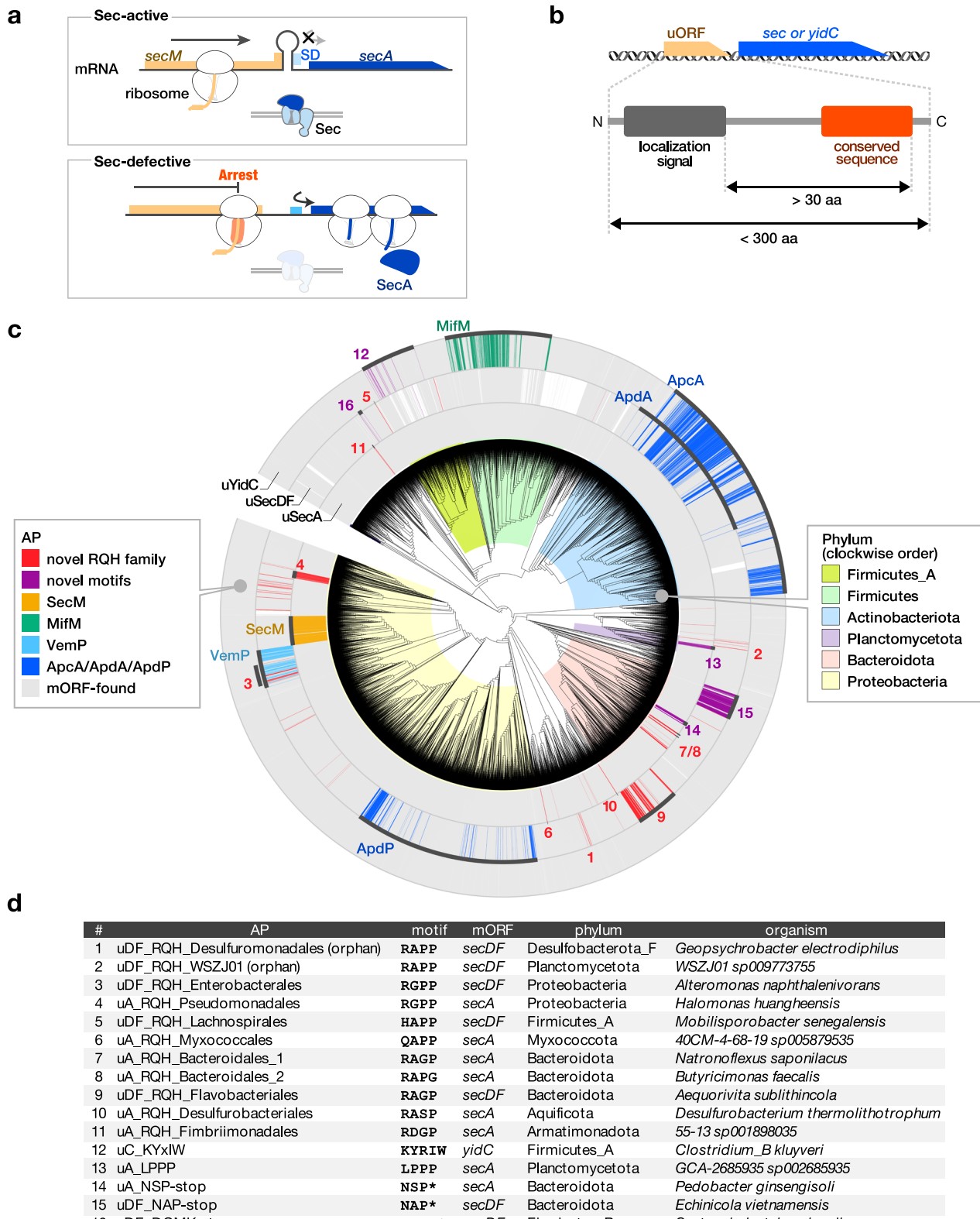

**Fig. 1 | Patchy and widespread phylogenetic distribution of candidate monitoring substrates across the bacterial tree of life. a** The translation arrest of SecM induces the *secA* gene by disrupting the stem-loop structure that otherwise sequesters the Shine–Dalgarno (SD) sequence. **b** Searching criteria for novel monitoring substrates. **c** Phylogenetic distribution of the candidate monitoring substrates. The gray strips around the bacterial phylogenetic tree indicate the genomes encoding SecA, SecD/F, or YidC homologs. Genomes with genes encoding putative arrest peptides located upstream of the *secA*, *secDF*, or *yidC* gene are indicated by chromatic strips, with the names or numbers corresponding to those reported in the list shown in (**d**). The colors behind the tree indicate the representative bacterial phyla listed on the right. **d** General information of the representative candidate arrest peptides used in vivo and in vitro experiments. The numbers correspond to those in (**c**).

from actinobacteria and ApdP from Alphaproteobacteria[28]. *apcA* was located upstream of *yidC2*, whereas *apdA* and *apdP* were located upstream of *secDF2*. Notably, crucial residues located near arrest sites exhibited a sequence similarity: ApdA and ApdP harbored the R-A-P-P sequence, whereas ApcA harbored the R-A-P-G sequence, which was also reminiscent of the R-A-G-P sequence corresponding to the C-terminal part of the arrest motif of SecM from *E. coli*[22,28].

Since we found that the screening described above could potentially be a groundbreaking approach to the identification of novel arrest peptides, we envisioned that we could comprehensively identify arrest peptides and depict a domain-wide phylogenetic view by applying it to a larger and comprehensive set of genome databases.

In this study, we conducted a systematic search for monitoring substrates to understand bacterial domain-wide nature of the arrest sequence. We utilized over 30,000 representative bacterial genomes of Genome Taxonomy Database (GTDB)[29], to guarantee the comprehensiveness. Strikingly, our current screening led to the identification of a large number of homology groups encoded upstream of the *secA*, *secDF*, and *yidC* genes. Interestingly, many of the ORFs identified bore RAPP-like sequences, such as RAPP, RGPP, RAGP, and RAGP. Furthermore, we identified several ORFs encoding distinct conserved sequences near the C-termini. Our subsequent in vivo and in vitro analyses provided evidence that they were able to efficiently stall the ribosomes of either *E. coli* or *B. subtilis*, or both. We also demonstrated that one of the arrest peptides identified here induced the downstream *secDF* gene in a translation arrest-dependent manner. These data suggest that a wide variety of bacteria have evolved arrest peptides encoded by uORFs of the *sec* or *yidC* genes to regulate these genes.

## Results

### Bioinformatics search for novel arrest peptides

To address the extent to which the arrest peptide-mediated regulation of genes involved in the protein localization machinery is universal in bacteria, we first carried out an in-depth in silico screening of ORFs encoding novel translation arrest peptides using over 30,000 representative bacterial genomes (Supplementary Data 1) from the GTDB[29]. We used the following search criteria, which we employed previously to identify ORFs encoding arrest peptides (Fig. 1b)[28]: (i) a uORF of the *sec* or *yidC* gene encoding a small protein with no annotated function; (ii) a uORF encoding an N-terminal secretion signal or a transmembrane (TM) sequence; (iii) a uORF encoding a C-terminal sequence conserved among its homologs; and (iv) a uORF encoding a spacer region between the localization signal and the C-terminal end of the conserved sequence with a size greater than 30 amino acid residues to ensure the exposure of the N-terminal localization signal outside of the ribosome when arrested. We extracted short ORFs of unknown function located upstream of the *secA*, *secDF*, and *yidC* genes and classified them into groups via clustering based on their amino acid sequences using the MMseqs2 function[30] or motif search (see Methods and Supplementary Methods). Subsequently, we collected candidate clusters or groups that met the criteria described above.

This in silico screening allowed us to identify several homologs of known arrest peptides, as well as many candidate ORFs that possibly encode novel arrest peptides (Fig. 1c, Supplementary Figs. 1–12, Supplementary Data 2–4). Strikingly, a substantial number of uORFs that met our criteria encompassed C-terminal motifs that were similar to that of SecM (R-A-G-P), ApcA (R-A-P-G), or ApdA/ApdP (R-A-P-P). We also identified uORFs that encode similar C-terminal motifs, such as R-G-P-P, H-A-P-P, and Q-A-P-P. In this study, we refer to these uORFs (i.e., those sharing the RAPP-like sequence that were not homologs of SecM, ApcA, ApdA, or ApdP) as the RQH family, as per the first residues of the consensus motifs. The RQH family members were widely, but also patchily, distributed among nine independent phyla (Fig. 1c, Supplementary Fig. 1). This contrasted with the ubiquitously distributed *secA*, *secDF*, and *yidC* genes (Fig. 1c, gray bars). The lack of an overall

sequence similarity among the RQH family members hampered their rational categorization into homology groups using conventional means. Therefore, we divided them into 19 groups based on the bacterial order in which they were identified, and provisionally named each of them based on the downstream gene, motif code (RQH, in this case), and bacterial order name using the following rule: "[uA/uDF/uC, which indicate a uORF of *secA*, *secDF*, and *yidC*, respectively]_[representative residues]_[bacterial order used in GTDB release 202]." For instance, members of the RQH family identified upstream of *secA* in a subset of the order Pseudomonadales are referred to as uA_RQH_Pseudomonadales (Fig. 1d). Among the 19 groups of the RQH family, eight groups were detected upstream of *secA*, whereas the remaining 11 groups were observed upstream of *secDF* (Supplementary Fig. 1a). In addition, we identified several uORFs that harbored the RAPP/RGPP motif but were apparently not conserved among their respective bacterial orders (less than three ORFs in each order; Supplementary Fig. 1b, c), possibly because of their too-narrow phylogenetic distributions. These orphan uORFs with RAPP/RGPP motifs were detected in nine phyla. Thus, for further in vivo and in vitro analyses, we selected nine representative uORFs from eight major RQH family groups, as well as two orphan uORFs (Fig. 1c, d).

Furthermore, we identified several clusters of uORFs that encoded a unique consensus motif apparently unrelated to that of any known arrest peptides. These uORFs were also provisionally named according to their downstream gene and conserved motif ([uA/uDF/uC]_[conserved residues]), as exemplified by uA_LPPP, which shares the L-P-P-P motif near the C-terminus (Fig. 1c, d, Supplementary Fig. 7). The members of the uC_KYxIW cluster detected in the Clostridia class of Firmicutes_A typically shared a unique K-Y-x-I-W sequence (x = less-conserved residue) (Fig. 1c, d, Supplementary Fig. 3). Several uORF clusters that shared a proline at the C-terminal end were identified exclusively in Bacteroidota (Fig. 1c, d, Supplementary Fig. 6). Members of the uA_NSP-stop and uDF_NAP-stop clusters encoded peptides encompassing N-S-P and N-A-P motifs, respectively, at their C-terminal ends. (Fig. 1c, d, Supplementary Fig. 6). Finally, uDF_DGMK-stop, which encodes D-G-M-K motif at the C-terminal end, was observed in subsets of Firmicutes_A and Firmicutes_B (Fig. 1c, d, Supplementary Fig. 3).

### In vitro translation arrest of the RQH family members

To address whether these candidate arrest peptides stall the ribosome, we did in vitro translation assay using the PURE system, a coupled in vitro transcription–translation system with all purified components derived from *E. coli* (*Ec* PURE)[31], as well as the *Bs* hybrid PURE system (*Bs* PURE)[32], in which only the ribosome of *Ec* PURE is replaced by the *B. subtilis* ribosome. For the in vitro translation assay, a gene segment encoding the C-terminal soluble region of a candidate arrest peptide (*ap*) was sandwich-fused between the *gfp* and *lacZα* genes (Fig. 2a). Ribosome stalling would result in the accumulation of the N-terminal GFP-AP fragment with a covalently bonded tRNA at the C-terminal end, which would migrate even slower than the full-length GFP-AP-LacZα fragment on SDS–PAGE, unless the tRNA moiety is removed by RNase pre-electrophoresis treatment.

Strikingly, all of the candidate arrest peptides tested here stalled the ribosome in the *Bs* PURE system; moreover, many candidate arrest peptides also arrested translation in the *Ec* PURE system (Fig. 2b–k, Supplementary Fig. 13). For instance, translation of the reporter for uDF_RQH_Desulfuromonadales in *Ec* PURE resulted in the accumulation of a major translation product of ~50 kDa that was reactive to an anti-GFP, but not to an anti-LacZα antibody (Fig. 2b, upper panels, lanes 1, 7). The RNase pre-treatment resulted in a mobility shift (Fig. 2b, upper panel, lane 2), indicating that this was an arrested product. A minor full-length product of ~43.5 kDa was also detected (Fig. 2b, upper panels, lanes 1, 2). The fraction arrest peptide ($f_{AP}$), which indicates the proportion of the arrest product among the total translation products was 0.96, suggesting that almost all of the

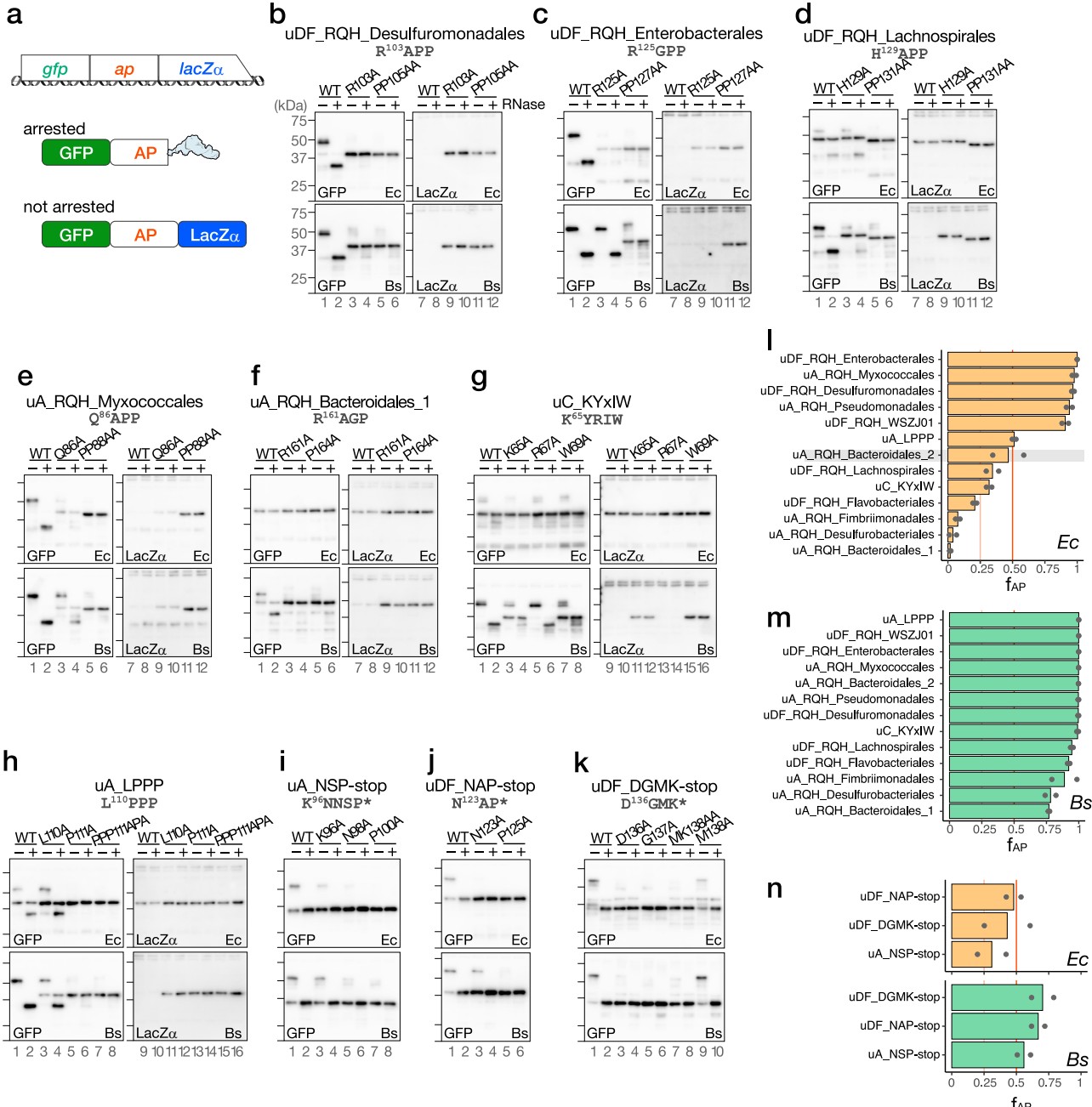

**Fig. 2 | In vitro analysis of translation arrest. a** Schematic representation of the *gfp-ap-lacZα* translational fusion template used for in vitro coupled transcription/translation and its translation products. "ap" indicates a gene segment encoding the C-terminal soluble region of each candidate arrest peptide. **b–k** Western blot analysis of the in vitro translation products. The reporter genes harboring wild-type (WT) or mutant derivatives of the putative arrest peptides indicated at the top of the figure were translated in the *E. coli* (*Ec*) and *B. subtilis* (*Bs*) PURE systems. The products were separated in neutral-pH gels and immunoblotted using anti-GFP (left) or anti-LacZα (right) antibodies. Before the separation, a portion of the samples were treated with RNase A (lanes indicated as +), to degrade the tRNA

translating ribosomes were stalled (Fig. 2l). The replacement of conserved arginine (Arg$_{103}$) or proline (Pro$_{105}$-Pro) residues by alanine residue(s) led to the predominant accumulation of the full-length product (Fig. 2b, upper panels, R103A, PP105AA). Similar results were obtained for translation using the *Bs* PURE system (Fig. 2b, lower panels), which yielded an $f_{AP}$ value of 1 (Fig. 2m). Other RQH family members, i.e., uDF_RQH_Enterobacterales, uA_RQH_Myxococcales,

moiety. Molecular size standards are indicated as horizontal lines on the left of each membrane; from top to bottom: 75, 50, 37, and 25 kDa, respectively. Experiments were conducted twice independently, with similar results. **l–n** Fraction arrest peptide ($f_{AP}$) of each reporter calculated according to the following formula: (arrest product) / (total translation products). The means of two independent experiments are indicated by the orange and green bars, which correspond to those obtained from experiments using the *Ec* (orange) or *Bs* (green) PURE systems, respectively. The dots indicate individual data points. Note that the stalling of uA_RQH_Bacteroidales_2 (shaded) may occur at a position other than the RAPG motif (see the main text). Source data are provided as a Source Data file.

uDF_RQH_WSZJ01, and uA_RQH_Pseudomonadales, which carry R$_{125}$GPP, Q$_{86}$APP, R$_{104}$APP, and R$_{109}$GPP sequences, respectively (Fig. 2c, e, Supplementary Fig. 13a, b), arrested translation in both *Ec* and *Bs* PURE, with $f_{AP}$ > 0.9 (Fig. 2l, m).

The remaining six members of the RQH family (uDF_RQH_Lachnospirales, uA_RQH_Bacteroidales_1, uA_RQH_Bacteroidales_2, uDF_RQH_Flavobacteriales, uA_RQH_Desulfurobacteriales, and

uA_RQH_Fimbriimonadales) arrested translation to lesser extents when translated using the *Ec* PURE system, whereas they efficiently arrested translation in the *Bs* PURE system (Fig. 2d, f, Supplementary Fig. 13c–f). Given that only the ribosome is different between *Ec* and *Bs* PURE, the difference in the arrest efficiencies between these two systems should be attributed to the difference in the ribosome structure. Although uA_RQH_Bacteroidales_2 exhibited an $f_{AP}$ value of 0.47 in the *Ec* PURE system (Fig. 2l), our toeprinting failed to identify the ribosome-stalling signal at the RAPG sequence (see below), which raised the possibility that the translation arrested product detected in the *Ec* PURE system was produced by ribosome stalling at a site other than the RAPG sequence. In accordance with this notion, neither the R102A nor the P104A mutation significantly abolished the translation arrest in *Ec* PURE, whereas they did so in *Bs* PURE (Supplementary Fig. 13c).

In most cases, alanine substitution of the conserved arginine or proline(s) in the RQH family members abolished or compromised translation arrest (Fig. 2b–f, Supplementary Fig. 13), thus highlighting the general importance of these conserved residues. Similarly, the Q86A substitution in the QAPP motif of uA_RQH_Myxococcales significantly reduced the efficiency of the translation arrest in both the *Ec* and *Bs* PURE systems (Fig. 2e). In turn, the H129A substitution in the HAPP sequence of uDF_RQH_Lachnospirales compromised translation arrest in *Bs* PURE, whereas the minor translation arrest observed in the *Ec* PURE system was unaffected by this same mutation (Fig. 2d). In contrast, the R125A mutation in uDF_RQH_Enterobacterales did not abolish the translation arrest in the *Bs* PURE system, whereas it did so in the *Ec* PURE system (Fig. 2c, R125A).

### Novel sequences that cause translation arrest in vitro

To examine the arrest capability of uORFs carrying C-terminal conserved motifs unrelated to known arrest peptides, we did an in vitro translation assay with arrest peptides with novel C-terminal motifs. Translation of the uC_KYxIW reporter using *Bs* but not *Ec* PURE resulted in the predominant accumulation of an RNase-sensitive arrest product (Fig. 2g, wild-type (WT)). The $f_{AP}$ values obtained using *Ec* and *Bs* PURE were 0.32 and 0.99, respectively (Fig. 2l, m). The arrest was impaired by the substitution of the highly conserved $Lys_{65}$ or $Trp_{69}$ residue with alanine (Fig. 2g, lower panels, K65A, W69A), but not by that of the less-conserved $Arg_{67}$ residue (Fig. 2g, lower panels, R67A).

A homolog of uA_LPPP also efficiently stalled *B. subtilis* ribosomes in vitro (Fig. 2h, lower panels), with an $f_{AP}$ value of 1 (Fig. 2m). The fraction of the arrested product was reduced when the $Leu_{110}$, $Pro_{111}$ or $Pro_{111}/Pro_{113}$ was substituted with alanine(s) (Fig. 2h, lower panels). In contrast, uA_LPPP only modestly stalled *E. coli* ribosomes (Fig. 2h, upper panels), with an $f_{AP}$ value of 0.51 (Fig. 2l). The mutations of proline residues but not $Lue_{110}$ decreased the accumulation of peptidyl-tRNA in *Ec* PURE (Fig. 2h, upper panels).

To evaluate the arrest efficiencies of the candidate arrest peptides uA_NSP-stop, uDF_NAP-stop, and uDF_DGMK-stop, we constructed a series of *gfp-ap* reporters in which a gene segment encoding the C-terminal soluble region and the subsequent stop codon of the candidate arrest peptide were fused after the *gfp* gene. The $f_{AP}$ was calculated based on the fraction of the peptidyl-tRNA among the total translation products. Approximately 30%–50% of the translation products of uA_NSP-stop, uDF_NAP-stop, and uDF_DGMK-stop were accumulated as a peptidyl-tRNA form when translated using *Ec* PURE (Fig. 2i–k, upper panels, WT, and 2n, upper graphs), whereas more than 50% of the translation products appeared as a peptidyl-tRNA form in *Bs* PURE (Fig. 2i–k, lower panels, WT, and 2n, lower graphs). In contrast, the direct fusion of the gene segment encoding the C-terminal region to *lacZa* led to the predominant accumulation of the full-length products (Supplementary Fig. 14), suggesting the importance of the termination codon for the arrest.

To examine the importance of the conserved amino acid residues near the C-terminus, we constructed mutant variants of the above

arrest peptides for an in vitro translation assay. The alanine substitutions of $Lys_{96}$, $Asn_{98}$, and $Pro_{100}$ in uA_NSP-stop reduced the peptidyl-tRNA accumulation in both *Ec* and *Bs* PURE, suggesting that the arrest depends on the amino acid residues or codons located at the −5, −3 and −1 positions from the C-terminal end (Fig. 2i, K96A, N98A, P100A). Essentially similar results were obtained for uDF_NAP-stop and uDF_DGMK-stop; the stalling was affected by mutations of the residues located at positions −3 and −1 in the case of uDF_NAP-stop, and −4, −3, −2, and −1 in the case of uDF_DGMK-stop (Fig. 2j, k).

### Determination of ribosomal-stalling sites using a toeprinting assay

To determine the ribosome stalling site, we performed a toeprinting assay, in which we employed a fragment analysis to determine the size of the toeprint product (Fig. 3a)[33–35]. A control experiment confirmed that the toeprint product generated by the SecM-stalled ribosome appeared as a single peak, with its size indicating that the ribosome stalled with the P-site at the $Gly_{165}$ (Fig. 3b, Supplementary Fig. 15), as demonstrated previously[34]. We then identified the stalling sites of the arrest peptides identified in this study using *Bs* PURE (Fig. 3c–f and Supplementary Figs. 16–31). For the arrest peptides that efficiently stalled both the *E. coli* and *B. subtilis* ribosomes, we also used the *E. coli* ribosome and confirmed that the estimated stalling site was identical. The results of our toeprinting analysis of the RQH family members revealed that, in all cases, the ribosome stalled when the codon for the third residue of the (R/Q/H)-(A/G/D)-(P/G/S)-(P/G) motif was in the P-site (Fig. 3c, d). Previous studies demonstrated that the stalling of ApcA, ApdA, and ApdP also occurred at the third position of the RAPG/RAPP sequences, suggesting that these arrest peptides share a common underlying mechanism[36].

Interestingly, our toeprinting analysis revealed that the ribosome stalling of uC_KYxIW occurred at a site located at a more C-terminal position than the conserved motif. The toeprinting of uC_KYxIW produced four consecutive toeprint signals (Supplementary Fig. 27). The strongest signal indicated that the ribosomes stalled at the $Phe_{75}$ codon in the P-site (Fig. 3e). Interestingly, the PTC-proximal residues were not well conserved among the homologs (Supplementary Fig. 3). The conserved and crucial $Lys_{65}$ and $Trp_{69}$ (Fig. 2g, Supplementary Fig. 3b) should be separated from the PTC by 11 and 7 residues, respectively, implying that they were situated in the mid-tunnel region in the NPET of the stalled ribosome. Further experiments will be necessary to address whether the minor signals indicate an additional stalling.

The toeprinting of uA_LPPP yielded two toeprint signals that indicated that the ribosome stalled when the P-site was at the third codon ($Pro_{112}$) of the $L_{110}PPP$ motif (Fig. 3e). Toeprinting of both uA_NSP-stop and uDF_NAP-stop yielded signals indicating that the stalling occurred when the A-site was at the stop codon (Fig. 3f, Supplementary Fig. 29, 30). In the case of uDF_DGMK-stop, the size of the major toeprint signal suggested that the stalling occurred either with the codon for $Lys_{139}$ or the stop codon located at the A-site (Supplementary Fig. 31). The importance of the stop codon for the arrest of uDF_DGMK-stop (Supplementary Fig. 14) renders it more likely that uDF_DGMK-stop induces the arrest when the stop codon is at the A-site (Fig. 3f).

### In vivo reporter assay to determine the efficiency of the translation arrest

To examine the efficiency of the elongation arrest in vivo, we did an in vivo reporter assay using the *gfp-ap-lacZ* in-frame fusion reporters (Fig. 4a). Elongation arrest before *lacZ* will result in a low β-galactosidase activity. We compared the β-galactosidase activity of WT arrest peptides with those of arrest-defective mutant variants, and calculated the translation arrest index (TAI), which is the ratio of the β-galactosidase activity of a mutant to that of WT (Fig. 4b–i,

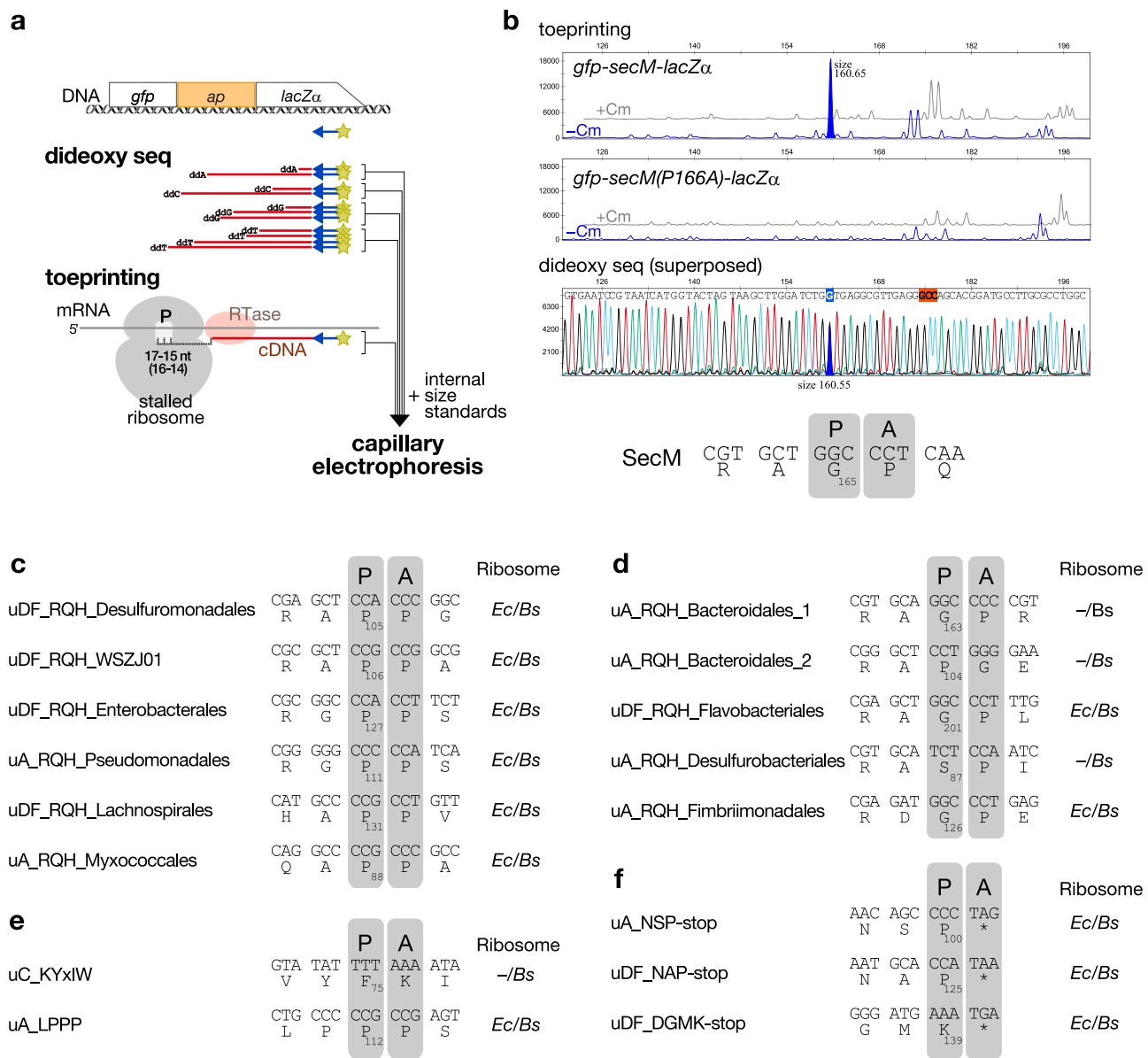

**Fig. 3 | Determination of the arrest sites using a toeprinting analysis.**
**a** Procedure used for toeprinting. **b** Toeprinting of SecM. The gray (+Cm) and blue (−Cm) lines in the top and middle panels indicate the signals obtained from experiments performed in the presence or absence of chloramphenicol, respectively. The stalling-dependent toeprint signal specifically obtained from the experiment using the wild-type, but not in that using the arrest-defective P166A mutant, is filled in blue. The peak corresponding to the stalling-specific toeprint signal and its nucleotide in the dideoxy sequencing data (bottom panel) are filled and marked in blue, respectively. Each fragment size calculated by the fragment

analysis is indicated beside those blue-filled peaks. The estimated codon in the P-site of the stalled ribosome is marked in red. The estimated stalling site is indicated as the codons in the P- (P) and A- (A) sites of the stalled ribosome and the P-site codon number. Additional details and raw plots are provided in Supplementary Fig. 15. (**c–f**), Estimated ribosome-stalling sites of the candidate monitoring substrates. The stalling sites determined for either or both *E. coli* (*Ec*) and *B. subtilis* (*Bs*) ribosomes are shown. "P" and "A" represent P-site and A-site positions, respectively. Raw plots are provided in Supplementary Fig. 16−31.

Supplementary Fig. 32). A high TAI value indicates an efficient translation arrest in vivo, whereas a TAI value as low as 1 suggests the absence of appreciable translation arrest in vivo. For example, *E. coli* cells harboring the reporter of uDF_RQH_Desulfuromonadales exhibited a low level of β-galactosidase activity (187 U), whereas the activities of the R103A and PP105AA mutant derivatives were 2,233 and 2,333 U, respectively (Fig. 4b). The TAI value calculated based on the β-galactosidase activities of the WT and PP105AA mutant strains was 12.51, which was indicative of an efficient translation arrest in *E. coli* (Fig. 4i). A similar result was obtained in the case of expression in *B. subtilis*, albeit with a relatively low TAI value of 3.18 (Fig. 4i).

When expressed in *E. coli*, most of the RQH family members yielded a high TAI value, with the exception of uA_RQH_Desulfurobacteriales and uA_RQH_Bacteroidales_2 (Fig. 4i). The former was consistent with its low arrest efficiency in *Ec* PURE. As mentioned above, the stalling of the latter in *Ec* PURE may have occurred at a site other than the RAPP-like sequence. Such an unrelated stalling may occur only in vitro. Low TAI values were obtained for uDF_RQH_Lachnospirales and uA_RQH_Desulfurobacteriales from experiments using *B. subtilis* (Fig. 4i). Nevertheless, the high TAI values obtained for most of the RQH family members in *E. coli* and *B. subtilis* demonstrated that they arrest translation efficiently in vivo.

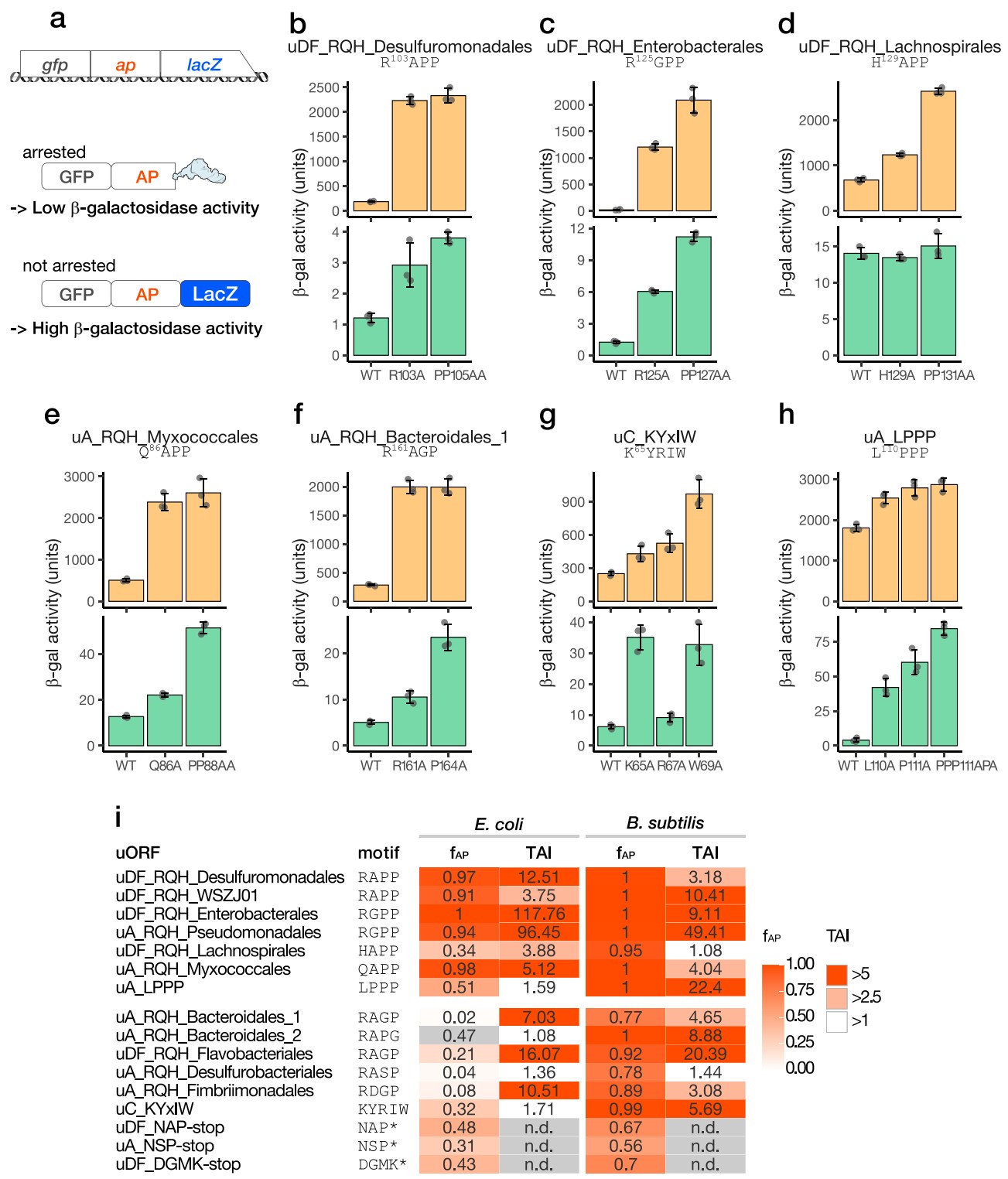

**Fig. 4 | In vivo translation arrest assay. a** Schematic representation of the reporter used for the in vivo assay and its translation products. **b–h** β-galactosidase activity (mean, *n* = 3 biologically independent cell cultures) of *E. coli* (orange bars) and *B. subtilis* (green bars) cells harboring wild-type (WT) or mutant derivatives of the arrest peptide reporters. The error bars and dots represent standard deviations and individual data points, respectively. **i** Summary of the in vitro and in vivo analyses of translation arrest. The translation arrest indexes (TAI) were calculated based on the in vivo β-galactosidase activities (mutant/wild-type) and are listed together with $f_{AP}$ (related to Fig. 2). Note that the stalling of uA_RQH_Bacteroidales_2 (gray) may occur at a position other than the RAPG motif (see the main text). Source data are provided as a Source Data file.

The TAI values of uA_LPPP in *E. coli* and *B. subtilis* were 1.59 and 22.4, respectively (Fig. 4h, i), suggesting that it efficiently stalls the *B. subtilis* but not *E. coli* ribosome in vivo. It is conceivable that the arrest of uA_LPPP observed in the *Ec* PURE system was an in vitro artifact caused by the absence of EF-P, which is required for efficient translation of poly-proline sequence[37–39]. The results of the in vivo analysis of uC_KYxIW showed a good agreement with those of its in vitro analysis, in which it stalled the *B. subtilis*, but not the *E. coli*, ribosome efficiently (Fig. 2h).

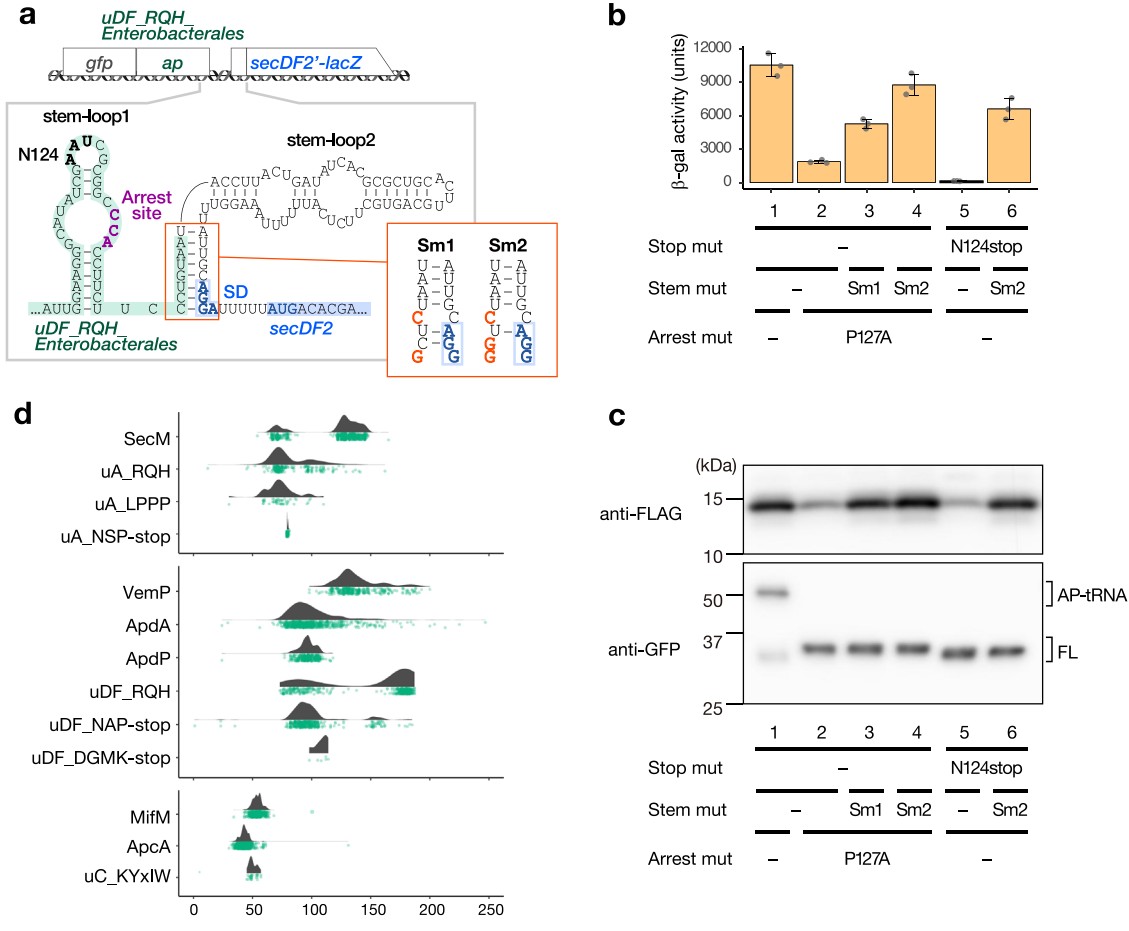

**Fig. 5 | Arrest-dependent regulation of downstream genes. a** Schematic representation of the *lacZ* reporter used to examine the expression of a downstream gene (upper) and the secondary structures of the intergenic region located between *uDF_RQH_Enterobacterales* (green) and *secDF2* (blue) (bottom). The translation arrest site is indicated in purple, and the N124 codon replaced by a stop codon within the mutant reporter in (**b**) and (**c**) is indicated in bold. The red panel shows the sequences of the Sm1 and Sm2 mutant reporters, in which the mutations introduced are indicated in bold red characters, whereas the putative the Shine–Dalgarno (SD) sequence of *secDF2* is indicated in bold blue characters. **b** In vivo analysis of *secDF2* regulation by uDF_RQH_Enterobacterales. Reporters harboring wild-type (lane 1) and a stop codon substitution mutant (lanes 5, 6), as well as an arrest-defective P127A mutant (lane 2) and its derivatives with stem-mutations (lanes 3, 4) of uDF_RQH_Enterobacterales, were expressed in *E. coli*, and the β-

galactosidase activities were measured (means ± standard deviations, *n* = 3 biologically independent cell cultures). **c** In vitro reconstitution of the arrest-dependent induction of the downstream gene. The gene encoding SecDF2′-LacZα−3xFLAG carrying the upstream *gfp-uDF_RQH_Enterobacterales* was translated in the *Ec* PURE system, and the translation products were analyzed by Western blotting using anti-FLAG (upper panel) and anti-GFP (lower panel) antibodies, respectively. Arrested (AP-tRNA) and full-length (FL) forms of the translation products are indicated. Experiments were conducted twice independently, with similar results. **d** Raincloud plots of the distances between the arrest sites (P-site residues) and the last residues of the N-terminal localization signal (distance_LS). The RQH family members encoded upstream of *secA* and *secDF* were combined and indicated as uA_RQH and uDF_RQH, respectively. The results of individual RQH family members are shown in Supplementary Fig. 33. Source data are provided as a Source Data file.

## The translation arrest of uDF_RQH_Enterobacterales induces the downstream secDF gene

The translation arrest of SecM, MifM, and VemP results in the induction of downstream target genes[9–11,24]. A similar role could be expected for the newly identified arrest peptides. To test this possibility, we focused on a homolog of uDF_RQH_Enterobacterales derived from *Alteromonas naphthalenivorans*, which belongs to the same order (Enterobacterales) as *E. coli* (Supplementary Fig. 2). We identified two predicted stem-loop structures, i.e., stem-loop 1 and stem-loop 2 (Fig. 5a), the latter of which partially masked the SD sequence of the downstream *secDF2* gene. We hypothesized that the stalled ribosome will disrupt stem-loop 2, keep the SD sequence exposed and thereby allow the expression of the downstream *secDF2* gene. To test this hypothesis, we constructed a reporter in which the coding region of GFP-uDF_RQH_Enterobacterales (residues 43–131) was followed by the intergenic region and the *secDF2′-lacZ* reporter (Fig. 5a).

An *E. coli* strain expressing the *secDF2′-lacZ* reporter described above exhibited a high β-galactosidase activity (10,531 units), which was drastically diminished by approximately 5.6-fold after the introduction of the P127A mutation in the $R_{125}$GPP motif of uDF_RQH_Enterobacterales (Fig. 5b, columns 1, 2). Abolishment of the arrest by the P127A mutation was confirmed using *Ec* PURE (Fig. 5c, lower panel, lanes 1, 2). These results suggest that the translation arrest of uDF_RQH_Enterobacterales strongly induces the expression of the downstream *secDF2* gene. The decreased induction of *secDF2* by the P127A mutation was partially counteracted by the disruption of two G-C base pairs (Sm1) or strongly counteracted by the disruption of three G-C base pairs (Sm2) in stem-loop 2 (Fig. 5a, b). These findings are consistent with the notion that the formation of the stem-loop 2 sequesters the SD sequence of the *secDF2* gene, thus repressing the induction of the *secDF2* gene. The premature translation termination that occurred after the introduction of a nonsense mutation at the 124th codon (Fig. 5a) resulted in an even lower β-galactosidase activity

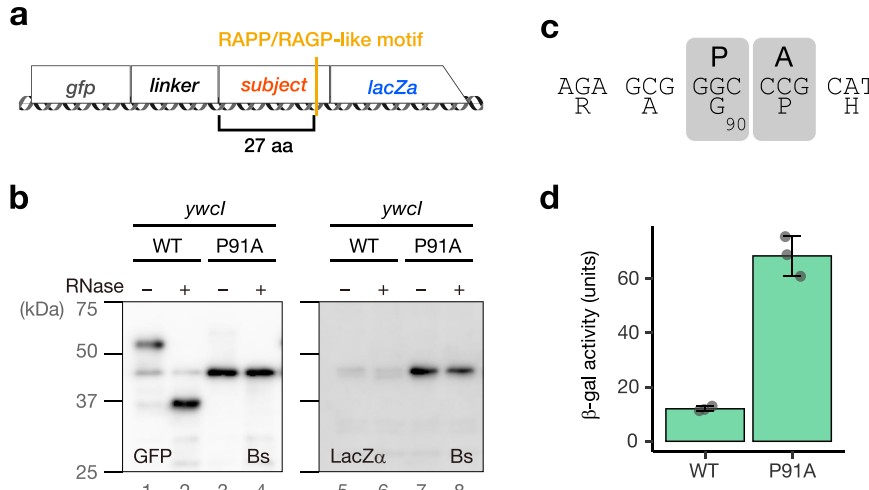

**Fig. 6 | Translation arrest by *B. subtilis* YwcI. a** Schematic representation of the *lacZα* reporter used for the in vitro translation of *E. coli* and *B. subtilis* proteins containing the RAPP/RAGP-like motif. The gene segments for target proteins containing RAPP-like sequences were sandwich-fused between *gfp* with a linker and *lacZα*. **b** In vitro analysis of the translation arrest of YwcI. Fusion genes harboring wild-type (WT) or the P91A mutant derivatives of *ywcI* were translated using the *Bs* PURE system. The products were separated on neutral-pH gels and immunoblotted using anti-GFP (left) or anti-LacZα (right) antibodies. Before the separation, a portion of the samples were treated with RNase A (lanes indicated as +), to degrade the tRNA moiety. Experiments were conducted twice independently, with similar results. **c** Estimated ribosome stalling site of *ywcI*. "P" and "A" represent the P-site and A-site positions, respectively. Raw plots are provided in Supplementary Fig. 35. **d** In vivo translation arrest analysis. β-galactosidase activity (means, *n* = 3 biologically independent cell cultures) of *B. subtilis* cells harboring wild-type (WT) or P91A mutant reporters. The error bars and dots represent standard deviations and individual data points, respectively. Source data are provided as a Source Data file.

compared with the basal activity observed for the P127A mutant; moreover, the abolishment of the induction was again counteracted by the Sm2 mutation (Fig. 5b, columns 5, 6). We assume that the ribosome that translates the P127A derivative can still transiently disrupt the stem-loop 2, which is completely eliminated by the premature translation termination by the N124stop mutation.

To further confirm that the translation arrest of uDF_RQH_Enterobacterales triggers the induction of the downstream gene, we did in vitro translation assay using *Ec* PURE. The arrest-dependent induction of the downstream gene was also recapitulated in vitro, in which the translation arrest of the GFP-uDF_RQH_Enterobacterales derivatives (Fig. 5c, lower) and induction of its downstream gene (*secDF2-lacZα–3xFLAG* reporter; Fig. 5c, upper) were assessed by anti-GFP and anti-FLAG antibodies, respectively. Taken together, these data support the notion that the translation arrest of uDF_RQH_Enterobacterales is responsible for *secDF2* expression via a mechanism similar to that of SecM, MifM, and VemP.

### Bioinformatics analysis of the length of the spacer between the protein localization signal and the arrest site

For an arrest peptide to function as a Sec- or YidC-monitoring substrate, the translation arrest must be released in a secretion- or membrane-insertion-dependent manner[27]. An optimal distance between the N-terminal TM segment and the arrest site is crucial for the membrane-insertion-dependent arrest cancellation if the TM segment adopts the type I (N-out/C-in) orientation[40,41], which is a topology that is often observed for YidC substrates[42]. Conversely, this distance can vary without impairing the localization-dependent arrest release if the N-terminal localization signal is either the Sec-dependent secretion signal or type II (N-in/C-out) TM segment[41]. We envisioned that this trend might be shared by known monitoring substrates, i.e., SecM, MifM, and VemP, as well as by other arrest peptides if they actually function as monitoring substrates.

To test these possibilities, we first determined the distance between the putative N-terminal localization signal and the arrest site of each homolog of SecM and plotted them to generate a distribution diagram (Fig. 5d, upper). We identified two discrete peaks with median distances of 131.5 (494 genomes) and 71 (145 genomes),

respectively, suggesting that SecM homologs can be divided into two classes, i.e., long and short variants, as reported previously[43]. In contrast, VemP, which is another known Sec-monitoring substrate, exhibited relatively long spacer lengths between the N-terminal localization signal and the arrest site, with a median distance of 133 (Fig. 5d, middle). In contrast, MifM, which is a YidC-monitoring substrate, exhibited a relatively narrow distribution pattern, with a shorter median distance of 54. These observations were consistent with our assumption that the spacer length of the YidC-monitoring substrate is relatively shorter and less variable than that of the Sec-monitoring substrate.

To investigate whether a similar trend will be observed for newly identified arrest peptides, we performed a similar analysis for ApcA, ApdA, ApdP, and arrest peptides identified in this study. Our analysis revealed that the arrest peptides encoded by the uORFs of *secA* (Fig. 5d, upper, Supplementary Fig. 33a) or *secDF* (Fig. 5d, middle, Supplementary Fig. 33b) typically had relatively longer spacer regions, with a wide range of length variations observed in most cases compared with MifM, ApcA, and uY_KyxIW, which are encoded by the uORF of *yidC* (Fig. 5d, lower). These observations suggest that these arrest peptides also share the molecular feature expected of monitoring substrates.

### Translation arrest of RAPP/RAGP motif-containing proteins in E. coli and B. subtilis

Previous and current studies revealed that many of the arrest peptides encoded by the uORFs of the *sec* or *yidC* gene harbor RAPP/RAGP-like motifs. We envisioned that other RAPP-containing proteins encoded by ORFs unrelated to the *sec* or *yidC* gene might also stall the ribosome. To test this possibility, we searched for proteins containing RAPP, RGPP, HAPP, HGPP, QAPP, QGPP, RAGP, RAPG, and RPPP sequences in the *E. coli* and *B. subtilis* proteomes (Supplementary Data 5). Among them, we chose 14 and 9 RAPP-like sequences derived from *E. coli* and *B. subtilis*, respectively, to examine their capability of stalling the ribosomes in vitro. The gene fragment encoding the RAPP-like motif was cloned between the *gfp* and *lacZ* genes (Fig. 6a). Translation using *Ec* or *Bs* PURE revealed that most of them did not arrest translation efficiently (Supplementary Fig. 34). However, we found that a

sequence derived from *B. subtilis* YwcI, i.e., a short peptide of 100 amino acids with an RAGP motif at residues 88–91, efficiently induced translation arrest in vitro (Fig. 6b), with an $f_{AP}$ value of 0.95. This arrest was abolished by replacing Pro$_{91}$ with alanine (Fig. 6b, P91A), suggesting that the proline of the RAGP motif is crucial for the arrest. A toeprinting analysis demonstrated that the ribosome stalled with the P-site at the Gly$_{90}$ codon (Fig. 6c, Supplementary Fig. 35). Finally, an in vivo reporter assay revealed a low level of β-galactosidase activity for the WT *ywcI* reporter, which was elevated after the introduction of the arrest-defective P91A mutation, resulting in a TAI value of 5.7 (Fig. 6d). These results demonstrated that YwcI is a novel arrest peptide derived from *B. subtilis*.

## Discussion

Our extensive screening led to the identification of dozens of arrest peptides that could possibly serve as Sec or YidC-monitoring substrates. Those include more than 10 members of the RQH family, which bear motifs similar to one another and to some of the arrest peptides identified previously, i.e., SecM, ApcA, ApdA, and ApdP. Other arrest peptides that bear novel arrest-inducing sequences were also identified. Our results highlight a unique pattern of evolution of the bacterial arrest peptides encoded upstream of the *sec* or *yidC* genes, which likely have emerged repeatedly in various bacterial species, resulting in a patchy and widespread phylogenetic distribution.

Arrest peptides often stall the ribosome in a species-specific manner, and, consistent with this notion, they generally have a narrow phylogenetic distribution. Therefore, the occurrence of the RQH family members in various bacterial phyla is an unusual evolutionary trait that has not been observed for other arrest peptides. Furthermore, we identified YwcI, an RAGP-containing arrest peptide encoded by a gene upstream of *sacT* (Fig. 6). SacT is an anti-transcriptional terminator of the *sacP* and *sacA* genes, which encode a sucrase and sucrose-specific permease, respectively[44–46]. This result suggests that the function of RAPP-containing arrest peptides is not limited to the regulation of the Sec or YidC pathway. In accordance with this notion, recent studies have suggested that the CruR and CutF uORFs, which harbor RAPP and poly-proline sequences, respectively, play roles in regulating the downstream ORFs encoding the TonB-dependent transporter BfrG and multicopper oxidase CutO, respectively[47,48], although the translation arrest of either of them has yet to be demonstrated experimentally.

The sharing of a similar RAPP-like sequence makes it plausible that these arrest peptides employ a similar nascent chain-ribosome interaction near the PTC to achieve ribosome stalling, as suggested by structural studies of ApdA, ApdP, and SecM[36,49]. This might also be the case for the arrest-inducing sequences containing RAPP-like motifs that were identified in a screening using a random sequence library[50,51]. However, the stalling efficiencies on *E. coli* and *B. subtilis* ribosomes varied among each arrest peptide (Figs. 2, 4), suggesting that the residues located upstream of the RAPP motif play an important role in determining the species specificity, as suggested for ApdA, and ApdP[28,36]. In addition, during the identification of YwcI, we found that most of the *E. coli* and *B. subtilis* proteins that contained RAPP-like motifs did not stall the ribosome (Supplementary Fig. 34). These observations point to the importance of the N-terminal region in the stalling ability. The N-terminal region may interact with the mid-tunnel region, in which ribosomal proteins, uL4, and uL22, as well as ribosomal RNA components sometimes encompass species-specific structures[52,53]. For example, the sequence of the internal region of uL22 that forms a constricted site in the NPET differs between *E. coli* and *B. subtilis*[52]. The difference in the uL22 was shown to be responsible for the species-specific ribosomal stalling of *B. subtilis* MifM[52]. Such species-specific structures in the NPET might have resulted in diverse and, sometimes, species-specific N-terminal sequences.

A key question arising from the current results is why the RAPP-containing arrest peptide alone was prevalent across the bacterial domain. One possibility is that the RAPP-like sequence serves as a versatile "seed" to tailor-make arrest-inducing sequences that can be flexibly optimized beyond the species-specific structural differences of each ribosome. The lack of an appreciable sequence similarity among the N-terminal region of the RAPP-containing arrest peptides suggests that a wide variety of N-terminal sequences might be compatible with the RAPP-like sequence without disrupting the ribosome-stalling capability, as suggested for SecM[54].

Another intriguing question is whether all of the arrest peptides that contained the RAPP-like sequence were derived from a common evolutionary origin or emerged repeatedly in different species and evolved independently. Although it is difficult to rule out one of these two possibilities, we favor the latter scenario for the following reasons: (i) these arrest peptides lacked overall sequence similarity to one another, with the exception of the RAPP-like motif; (ii) gene contexts were divergent among these arrest peptides; and (iii) the repetitive occurrence of such a short sequence motif during evolution seems possible.

We also successfully identified arrest peptides that bore novel arrest motifs. Those included uC_KYxIW, uA_LPPP, uA_NSP-stop, uDF_NAP-stop, and uDF_DGMK-stop. Among them, uC_KYxIW was unique in that its conserved and arrest-essential motif was located 7–11 residues distal from the stalling site and the PTC-proximal residues were less conserved (Supplementary Fig. 3). The identification of various distinct arrest sequences suggests that bacteria have evolved various arrest-inducing mechanisms and that there must be more unidentified arrest-inducing sequences.

In the present study, the results of the in vivo and in vitro translation arrest assays were generally in great agreement with each other. However, inconsistencies remained in some cases. For instance, uDF_RQH_Desulfuromonadales, uDF_RQH_Lachnospirales, uA_RQH_Myxococcales, uA_RQH_Desulfurobacteriales, and uA_RQH_Fimbriimonadales stalled the *B. subtilis* ribosomes efficiently in vitro but exhibited a relatively lower arrest efficiency in the in vivo experiment (Fig. 4i). The absence of EF-P in our in vitro translation systems may explain the discrepancies observed for the former three arrest peptides, which contains the di-proline sequence in their conserved motifs (RAPP, HAPP and QAPP, respectively). This might be the case for uA_LPPP, which stalled the *E. coli* ribosome only in vitro (Figs. 2 and 4) as mentioned. Other possible explanations for these results are that the ribosome stalling caused by the foreign peptides was somehow subjected to release by a cellular factor, resulting in a reduced stability of arrest in vivo or that the translation product of the reporter became a target of proteolysis in vivo, thus causing an apparent inconsistency. Note that the strains expressing uDF_RQH_Desulfuromonadales exhibited only low levels of β-galactosidase activity (Fig. 4b). Thus, it might possibly compromise the reliability of the quantified data. Conversely, uA_RQH_Fimbriimonadales efficiently stalled the *E. coli* ribosome in vivo but not in vitro. It is formally possible that an in vivo co-factor or specific condition is required for the stabilization of the arrest.

We demonstrated that the elongation arrest by *A. naphthalenivorans* uDF_RQH_Enterobacterales led to the induction of the downstream *secDF2* gene (Fig. 5a–c). The bioinformatics analysis of the spacer length between the localization signal and the arrest motif also suggests that the arrest peptides identified in this study share the molecular feature expected of monitoring substrates (Fig. 5d). These observations support the notion that the arrest peptides identified in this study function as monitoring substrates.

The arrest peptides encoded upstream of the *sec* genes had relatively longer spacer regions with length variations in general (Fig. 5d, Supplementary Fig. 33). Markedly, the spacer length of SecM homologs exhibited discrete bimodal distributions (Fig. 5d), as

reported previously[43,54]. The longer variants were found from bacteria of Enterobacterales, including *E. coli*, whereas the shorter variants were found from bacteria of Pasteurellales (family Pasteurellaceae of order Enterobacterales in GTDB release 202). A specific spacer length might be required to coordinate the timing of membrane targeting and synthesis of the arrest motif, as proposed previously[55]. If bacteria of different lineage require each unique optimum length, it might result in a bimodal length distribution. It is also worth noting that the spacer region could contain a regulatory element, as reported for *E. coli* SecM[56] or *V. alginolyticus* VemP[57]. Occurrence of an additional regulatory element within the spacer region during evolution might also result in the spacer length variation.

This study opened up the possibility that the search for proteins containing the RAPP-like sequence may allow the identification of novel arrest peptides, as demonstrated for *B. subtilis* YwcI. Further identification and characterization of novel arrest peptides will provide insights into the shared or lineage-specific evolution of arrest peptides, during which each bacterium must have achieved various physiological functions and mechanisms of translation regulation through common or unique interactions between the ribosome and the nascent peptide chains. The exploration of various regulatory arrest peptides will unveil unidentified principles via which the ribosome translates genetic information into cellular functions in a manner that is beyond our current understanding.

## Methods

All unique materials are available from the corresponding authors.

### In silico search for uORFs encoding putative monitoring substrate-like arrest peptides

Genomic identifiers (Refseq ID and Genbank ID) of representative bacterial genomes of the Genome Taxonomy Database[29] (GTDB, release 202) were collected and used to download files of genomic sequences, protein sequences, and GFF annotation files from the NCBI FTP server using ncbi-genome-download script (https://github.com/kblin/ncbi-genome-download). A total of 30,175 bacterial genomes were subjected to the in silico screening as described below (Supplementary Data 1). Putative YidC, SecA, and SecDF homologs were blast-p searched (E < 10^{-4}) from the local protein database made from downloaded protein sequences using BLAST + [58,59] version 2.13.0. Amino acid sequences of *E. coli* and *B. subtilis* homologs of YidC (NP_418161.1 [https://www.ncbi.nlm.nih.gov/protein/NP_418161.1/] for *E. coli* YidC and NP_391984.1 [https://www.ncbi.nlm.nih.gov/protein/NP_391984.1] for *B. subtilis* YidC), SecA (NP_414640.1 [https://www.ncbi.nlm.nih.gov/protein/NP_414640.1] for *E. coli* SecA and NP_391410.1 [https://www.ncbi.nlm.nih.gov/protein/NP_391410.1] for *B. subtilis* SecA), and SecDF (NP_414942.1 [https://www.ncbi.nlm.nih.gov/protein/NP_414942.1] for *E coli* SecD, NP_414943.1 [https://www.ncbi.nlm.nih.gov/protein/NP_414943.1] for *E. coli* SecF, and for NP_390643.1 [https://www.ncbi.nlm.nih.gov/protein/NP_390643.1] *B. subtilis* SecDF) were used as queries. Protein IDs of the putative YidC, SecA, and SecDF homologs thus collected were used to search for their GFF annotation files, which were then used to extract information about their upstream genes that are in the same orientation as the downstream *sec/yidC* genes. Subsequently, putative uORFs that encode small proteins (<300 aa) of either unknown function (protein products annotated as hypothetical, putative, uncharacterized, unknown, DUF, membrane protein, or extracytoplasmic protein) or homologs of known monitoring substrates (SecM, MifM, VemP, monitor, translation regulator, or regulator of OxaAB translation) were selected. Signal sequences and transmembrane (TM) regions within the uORF were predicted using SignalP[60] version 6.0, deepTMHMM[61] version 1.0.18, and TMHMM[62] version 2.0. The resulting datasets were subsequently used for clustering. During the above analysis, we utilized R scripts developed in-house (with R version 4.0 or later).

### uORF clustering by MMseqs2

To classify uORFs into homology groups, we used MMseqs2 software (version 14-7e284) that enables searching and clustering of huge sequence sets[30]. Before clustering each uORF of *secA*, *secDF*, and *yidC*, we first prepared non-redundant representative sequence databases by removing redundant sequences from the initial uORF datasets. We then obtained sequence profiles by the first search using each representative sequence database as a query against each representative sequence database. The subsequent second search was carried out using each profile database as a query against each profile database. The search results were used for clustering either by the default greedy set cover clustering algorithm (cluster_s1, --cluster-mode 0) or connected component algorithm (cluster_s2, --cluster-mode 1) that covers more remote homologs. Amino acid sequences in each cluster were aligned by MAFFT[63,64] version 7.490, and their conservation was visualized by sequence logo using ggseqlogo[65] version 0.1.

### Classification of SecM

We selected a cluster generated by cluster_s1 that contained WP_000014321.1 [https://www.ncbi.nlm.nih.gov/protein/WP_000014321.1] whose amino acid sequence was identical to that of SecM (NP_414639.2 [https://www.ncbi.nlm.nih.gov/protein/NP_414639.2]) of *E. coli* str. K-12 substr. MG1655 (GCF_000005845.2 [https://www.ncbi.nlm.nih.gov/datasets/genome/GCF_000005845.2/]). We aligned the 645 sequences included in the raw cluster using MAFFT FFT-NS-2 algorithm and removed a sequence that obviously lacked arrest motif represented by "IRAGP," identified previously as the critical residues for ribosome stalling[22]. GCF_001908105.1 had two chains (NZ_MKGQ01000011.1 [https://www.ncbi.nlm.nih.gov/nuccore/NZ_MKGQ01000011.1], NZ_MKGQ01000006.1 [https://www.ncbi.nlm.nih.gov/nuccore/NZ_MKGQ01000006.1]) that have identical *secM-secA* sequences (locus_tags: Xedl_RS29740-Xedl_RS29745 and Xedl_RS31575-Xedl_RS31580, respectively). GCF_008710095.1 [https://www.ncbi.nlm.nih.gov/datasets/genome/GCF_008710095.1/] had one chain (NZ_VYKJ01000001.1 [https://www.ncbi.nlm.nih.gov/nuccore/NZ_VYKJ01000001.1]) wherein two sets of an identical *secM-secA* sequence (FJU30_RS00920-FJU30_RS00915 and FJU30_RS03845-FJU30_RS03850) were present. We exclusively utilized one instance from each, correspondingly. After removing the sequences that were not predicted to bear the N-terminal signal sequence (n = 3), we obtained a non-redundant set of SecM homologs, including 626 sequences derived from 639 genomes.

### Classification of MifM

We selected a uORF cluster of *yidC*, generated by the cluster_s2, that contained MifM (YP_054586.1 [https://www.ncbi.nlm.nih.gov/protein/YP_054586.1]) of *B. subtilis* (GCF_000009045.1 [https://www.ncbi.nlm.nih.gov/datasets/genome/GCF_000009045.1/]).

Among the raw cluster containing 631 sequences, three sequences were derived from Actinobacteriota, six were from Firmicutes_B, and the remaining 619 were from Firmicutes. Many uORFs in the cluster, except for those derived from Actinobacteriota, shared a region rich in negatively charged residues that appears within the amino acid residues ranging from 75 to 95 in many cases. It has been demonstrated that the continuous negatively charged residues of *B. subtilis* MifM are critical for the ribosome stalling[35]. We calculated local negative charge using a sliding window of 5 amino acids using idpr[66] R package (version 1.8.0) and selected uORFs of Firmicutes whose minimum local charge in the five amino acid window is below −0.3 within the region between 70th and 100th residue. By removing the sequences that were not predicted to have the N-terminal localization signal or that have too short soluble domain (<30 aa) after the localization signal, we obtained non-redundant 572 protein sequences that are derived from 592 genomes of Firmicutes.

## Classification of VemP

We selected a cluster generated by cluster_s1 that contained VemP (WP_017819886.1 [https://www.ncbi.nlm.nih.gov/protein/516430831]) of *Vibrio alginolyticus* (GCF_000354175.2 [https://www.ncbi.nlm.nih.gov/datasets/genome/GCF_000354175.2/]). After the MAFFT FFT-NS-2 alignment of 369 sequences in the cluster, we removed four uORFs, in which the consensus RxxxWKxxNxxY-like motif was not (*n* = 1), or only partially (*n* = 2) conserved. Subsequently, 24 uORFs that lacked a putative localization signal were removed. Thus, we obtained non-redundant 339 VemP-like sequences derived from 339 genomes. We found that *Vibrio* sp. 10 N.286.49.B1 (GCF_002873335.1 [https://www.ncbi.nlm.nih.gov/datasets/genome/GCF_002873335.1/]) and *Vibrio hangzhouensis* (GCF_900107935.1 [https://www.ncbi.nlm.nih.gov/datasets/genome/ GCF_900107935.1/]) have two distinct *vemP-secD/F* copies in each genome.

## Classification of ApcA

A cluster containing ApcA (WP_029256089.1 [https://www.ncbi.nlm.nih.gov/protein/WP_029256089.1]) of the *Rhodococcus erythropolis* PR4 (GCF_000010105.1 [https://www.ncbi.nlm.nih.gov/datasets/genome/GCF_000010105.1/]) was selected. The cluster comprises 1,618 sequences derived only from the phylum Actinobacteriota. This cluster was generated by the cluster_s2 with a sensitivity value of 7, which was employed for the second search during the clustering process. We aligned them using MAFFT FFT-NS-2, and removed 17 sequences that did not have RAPx or RGPx motif. One sequence was further removed because it was not predicted to bear the N-terminal localization signal. *Streptomyces carminius* (GCF_002794255.1 [https://www.ncbi.nlm.nih.gov/datasets/genome/GCF_002794255.1/], Assembly level is "Scaffold") have two scaffolds (NZ_PGGW01000008.1 [https://www.ncbi.nlm.nih.gov/nuccore/NZ_PGGW01000008.1] and NZ_PGGW01000038.1 [https://www.ncbi.nlm.nih.gov/nuccore/NZ_PGGW01000038.1]) that have identical *apcA-yidC* sequences (locus_tags: CUT44_RS01795-CUT44_RS01800 and CUT44_RS10055-CUT44_RS10060, respectively). We exclusively utilized the former sequence for analysis. Thus, we obtained non-redundant 1,590 ApcA-like sequences derived from 1,599 genomes.

## Classification of uC_KYxIW

The group uC_KYxIW was identified as a cluster that met our criteria to search for putative monitoring substrates. The cluster, generated by cluster_s1, contained 26 sequences derived from the phylum Firmicutes_A. In addition to the 26 sequences, one more sequence was clustered together when the sensitivity of the second search during the clustering was increased from the default value 4 to 7. All 27 sequences have the N-terminal localization signals, among which 22 were predicted to have either the N-out/C-in single TM segment or N-in/C-in double TM segments with a small extracytoplasmic region, as has also been seen for MifM and ApcA. Thus, we obtained 27 uORF sequences derived from 27 genomes.

## Classification of uDF_NAP-stop, uA_NSP-stop, uDF_DGMK-stop

After clustering, we identified that six clusters of ORF located upstream of *secDF* shared a conserved proline residue at the C-terminal end. They were mainly found in subsets within three orders of the phylum Bacteroidota. Because only one sequence was derived from the phylum Proteobacteria, we discarded it. We grouped these clusters into a single family that we refer to as uD_NAP-stop. From the initial set of the uORFs within this class, we removed five redundant sequences and three uORFs, in which the N-terminal localization signal was not predicted. Thus, we obtained a dataset of uDF_NAP-stop, which comprises 396 uORF sequences derived from 400 genomes.

We also identified a uORF cluster of *secA* whose members share a conserved C-terminal Pro residue. The cluster comprises 66 sequences derived from the family Sphingobacteriaceae of the phylum Baceroidota. We discarded one sequence that had no predicted N-terminal localization signal. After removing redundant sequences, we finally obtained 63 non-redundant uA_NSP-stop sequences from 64 genomes.

The group uDF_DGMK-stop was found in a uORF cluster of *secDF*. This cluster comprises four and two sequences from Firmicutes_B and Firmicutes_A, respectively. According to the NCBI taxonomy, all of them are classified into the class Clostridia of the phylum Firmicutes. All sequences have predicted the N-terminal localization signals. Thus, we categorized them into the group uDF_DGMK-stop.

## Determination of query motifs for the motif search to identify the RQH family members

We found that the classification of ApdA, ApdP, and other members of the RQH family using clustering was challenging, likely due to the lack of overall sequence similarity except for the conserved RAPP-like sequence within each homology group. Therefore, in this study, we employed motif search to identify uORF encoding proteins bearing the RAPP-like sequence at the C-terminus. For searching queries, we chose motifs of known arrest sequences whose stalling activities were experimentally demonstrated (RAGP; *E. coli* SecM, RAPP; ApdA and ApdP, RAPG; ApcA, RGSP; *Mannheimia succiniciproducens* SecM). In addition, we chose motifs LAGP, RADP, and RASP, which were found in uORFs clustered with *E. coli* SecM, as well as motifs RAPG, RAPS, RAPA, RAPT, RAP*, RGPT, RAPQ, RAPE, RGPS, RGPG, RAPC, RAPD, and RAPV, that were found in uORFs clustered with ApcA. We also chose the RGPP, HGPP, and RSPP motifs, which were found in uORFs clustered with those having the RAPP sequence.

In addition to the above motifs, we chose the QAPP, HAPP, TGPP, and RDGP motifs as follows. We explored C-terminally conserved motifs that particularly contained X-A-P-P, X-G-P-P, or R-X-G-P sequences from short uORFs with N-terminal localization signal. If a uORF contained multiple di-proline motifs, we considered only the most C-terminal one. We then picked up motifs that were shared by at least four uORFs sequences derived from species in the same bacterial order. If the uORFs were clustered into a group, in which the C-terminal RAPP-like sequence was not conserved among the group members, we excluded these sequences. It allowed us to choose the QAPP, HAPP, TGPP, and RDGP motifs.

## Classification of ApdA, ApdP, and other RQH classes

We conducted a motif search using the aforementioned RAPP-like motifs as queries. We then selected uORFs if more than two uORFs that share a conserved query motif were found in species that belong to the same order. We then classified the selected uORFs into individual groups based on the bacterial order they belong to and termed the RQH family, except for previously identified ApdA and ApdP, which are conserved among species that belong to multiple bacterial orders. Consistent with our previous search showing that ApdA is derived from the phylum Actinobacteriota[28], our current motif search has identified 401 uORFs with RAPP-like motifs derived from a subset of Actinobacteriota, including the orders Streptomycetales, Streptosporangiales, Mycobacteriales, Propionibacteriales, Jiangellales, and others (Supplementary Fig. 4). We removed three sequences that lacked putative N-terminal localization signal and then classified the remaining 398 sequences into ApdA, which were derived from 398 genomes. Similarly, 293 uORFs with RAPP-like motifs found within Alphaproteobacteria were classified as ApdP. These were found within 292 genomes that belong to the orders Rhizobiales, Rhizobiales_A, Rhodobacterales, Azospirillales, Caulobacterales, and others (Supplementary Fig. 5). Two distinct *apdP-secDF* sets were found in the current version of *Nitratireductor soli* genome information (GCF_001050155.1 [https://www.ncbi.nlm.nih.gov/datasets/genome/GCF_001050155.1/]).

In addition to the above RQH family members, we also selected orphan ORFs by motif search using the RAPP and RGPP sequences as

queries. If only one or two ORFs that share a conserved query motif were found in species that belong to the same order, we selected them as orphan ORFs (Supplementary Fig. 1b).

**Classification of uA_LPPP**
Our cluster analysis identified a subset of uORFs that encoded proteins bearing a conserved LPPP motif within their C-terminal 15 amino acid residues in the orders Phycisphaerales, UBA1161, and WQYP01 of the phylum Planctomycetota. Those sequences were dispersed to multiple clusters by any clustering condition tested, presumably because of the diversity in their N-terminal sequences. Therefore, we did a motif search through the genomes derived from the above three orders to collect uORF sequences that have LPPP as well as XPPP motifs within the last 15 amino acids. We identified 52 sequences, of which 45 had LPPP motifs. We then selected 50 sequences that were predicted to have the N-terminal signal sequence.

**Phylogenetic tree.** Phylogenetic tree data were downloaded from the GTDB (release 202). If needed, the tree was split into a specific phylum using the "drop.tip" function of ape[67] version 5.6.2. The tree was visualized and decorated using iTol[68] version 6.

**Bacterial strains and plasmids.** The *B. subtilis* strains, plasmids, and DNA oligonucleotides used in this study are listed in Supplementary Data 6, 7, and 8, respectively. The preparation of synthetic DNAs for candidate monitoring substrates was outsourced (Thermo Fisher). Plasmids were constructed via standard cloning methods, including PCR using PrimeSTAR GXL (Takara), and DpnI treatment (Takara); moreover, Gibson assembly[69]. Sera-Mag Carboxylate-Modified Magnetic Particles (Cytiva, 65152105050250) was used to purify double-stranded DNA[70] and sequencing products[71]. The *B. subtilis* strains were constructed by transformation involving double homologous recombination between chromosomal DNA and the plasmids introduced into *B. subtilis* competent cells. The resulting recombinant clones were validated based on their antibiotic-resistance markers.

**Culture media and growth conditions.** *B. subtilis* cells were cultured in LB medium. *E. coli* cells were cultured in LB medium supplemented with 100 µg/ml ampicillin. Cells were cultured at 37 °C and collected for Western blotting or β-galactosidase activity assay when they reached an optical density of 0.5–1.0 at 600 nm ($OD_{600}$).

**Antiserum production.** The production of the anti-LacZα antiserum was outsourced to Eurofins. Two chemically synthesized peptides, NH2-CRNSEEARTDRPSQQ-COOH and NH2-CTDRPSQQLRSLNGE-COOH, corresponding to residues 38–51 and 45–58 of *E. coli* LacZ, respectively, were used for immunization. N-terminal cysteines were added to both polypeptides, for their conjugation to the carrier protein Keyhole limpet hemocyanin.

**In vitro translation and Western blotting.** Bacterial reconstituted transcription–translation coupling systems[31,32] were used in the in vitro translation assay. Specifically, for in vitro translation using *E. coli* ribosomes, we utilized PUREfrex version 1.0 (GeneFrontier) according to the manufacturer's protocol. For *Bs* PURE, purified *B. subtilis* ribosomes were used at the final concentration of 1 µM in the PURE system, without adding *E. coli* ribosomes. Then, 2.5 U/µL of T7 RNA polymerase (Takara) was added, to ensure transcription. The in vitro translation reaction was primed using the DNA templates listed in Supplementary Data 9. The translation reaction was carried out for 30 min at 37 °C and was stopped by adding 2× SDS–PAGE loading buffer, for Western blotting. A portion of the sample was further treated with 0.2 mg/ml RNase A (Promega) at 37 °C for 10 min, to degrade the tRNA moiety of the peptidyl-tRNA, if necessary. The translation products were separated on a 10% polyacrylamide gel that was prepared using WIDE

RANGE Gel buffer (Nacalai Tesque), according to the manufacturer's instructions, then transferred onto a PVDF membrane (Merck, IPVH00010) and subjected to immuno-detection using antibodies against GFP (mFX75; Wako) or LacZα. Anti-Mouse IgG (BIO-RAD, 170-6516) and anti-Rabbit IgG (BIO-RAD, 170-6515) were used as the secondary antibodies for anti-GFP and anti-LacZα, respectively. The primary and secondary antibodies were diluted 1:5000 for use. Bands were visualized by using ECL Prime Western Blotting Detection Reagent (Cytiva, RPN2236) with an Amersham Imager 600 (GE Healthcare), and the band intensities were quantified using Image-Quant TL (GE Healthcare).

**Toeprinting assay.** In vitro translation was carried out using the *Ec* PURE or *Bs* PURE system at 37 °C for 20 min in the presence or absence of 0.1 mg/mL chloramphenicol, a translation inhibitor. The translation reaction mixture was then mixed with the same volume of the reverse transcription mixture containing 50 mM HEPES-KOH, pH 7.6, 100 mM potassium glutamate, 2 mM spermidine, 13 mM magnesium acetate, 1 mM DTT, 2 µM of oligonucleotide labeled with 6-carboxyfluorescein (6-FAM) at the 5′ end (5′-AACGACGGCCA GTGAATCCGTAATCATGGT-3′, Invitrogen), 50 µM each dNTP, and 10 U/µL ReverTra Ace (Toyobo), then incubated further at 37 °C for 15 min. The reaction mixture was diluted 5-fold with the NTC buffer (Macherey-Nagel), and the reverse transcription products were purified using a NucleoSpin Gel and PCR Clean-up kit (Macherey-Nagel). The reverse transcription products were eluted with 30 µL of HiDi formamide (Thermo Fisher). Samples were then mixed with 10 µL of 10-fold-diluted GeneScan 500 LIZ dye size standard (Thermo Fisher, 4322682), then heated at 96 °C for 3 min just before capillary electrophoresis. The dideoxy DNA samples used as size markers for sequencing were prepared using a Thermo Sequenase Dye Primer Manual Cycle Sequencing Kit (Thermo, 79260), Thermo Sequenase Cycle Sequencing Kit (Thermo, 785001KT), or Thermo Sequenase DNA Polymerase (Cytiva, E79000Y), according to the manufacturer's instruction, with some modifications. The DNA polymerase reaction was carried out using the same sets of template DNA and primer used for the toeprint assay. Each reaction mixture contained 0.44 µM of the 6-FAM-labeled primer, 60 µM each deoxynucleotide triphosphate (dATP, dCTP, dGTP, and dUTP), and 0.6 µM dideoxynucleotide triphosphate (either ddATP, ddCTP, ddGTP, or ddUTP). The sequencing products were purified using Sera-Mag speed beads and eluted with HiDi formamide. Next, 2 µL of a 10-fold-diluted GeneScan 500 LIZ dye size standard was added. If needed, the toeprinting product was further diluted before electrophoresis using HiDi formamide. The toeprinting and dideoxy sequencing products were then subjected to fragment analysis on a Seqstudio genetic analyzer (Thermo Fisher). Fragment data were analyzed and visualized using the GeneMapper software version 6 (Applied Biosystems), and processed further using Adobe Illustrator. The signals obtained from dideoxy sequencing were colored green (A), blue (C), black (G), and red (T), and then superposed for presentation (Fig. 3b and Supplementary Fig. 15).

**In vivo β-galactosidase assay.** The β-galactosidase assay was performed as described previously[6]. A 100-µL aliquot of the culture was transferred to a well in a 96-well plate, and $OD_{600}$ was recorded. We mixed the culture with 50 µL of Y-PER reagent (Thermo Fisher) for 20 min at room temperature, to disrupt the cells. In the case of *E. coli* cells, the mixture was diluted 10-fold and further subjected to freeze–thaw treatment, to ensure cell disruption. Subsequently, 30 µL of *O*-nitrophenyl-β-D-galactopyranoside (ONPG) in Z-buffer (60 mM $Na_2HPO_4$, 40 mM $NaH_2PO_4$, 10 mM KCl, 1 mM $MgSO_4$, and 38 mM β-mercaptoethanol) was added to the cell lysate, and the $OD_{420}$ and $OD_{550}$ were measured at 28 °C every 5 min over a period of 60 min. Arbitrary units of β-galactosidase activity were calculated using the following formula: $[(1000 \times V_{420} - 1.3 \times V_{550}) / OD_{600}]$ for *B. subtilis*, or

$[10 \times (1000 \times V_{420} - 1.3 \times V_{550}) / OD_{600}]$ for *E. coli*; where $V_{420}$ and $V_{550}$ are the first-order rate constants, $OD_{420}/min$ and $OD_{550}/min$, respectively.

## Reporting summary

Further information on research design is available in the Nature Portfolio Reporting Summary linked to this article.

## Data availability

Source data are provided with this paper.

## Code availability

Custom R codes for the bioinformatic search and data analyses are provided as Supplementary Software.

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

## Acknowledgements

We thank Machiko Murata and Naoko Muraki for their technical support. This work was supported by JSPS Grant-in-Aid for Scientific Research (Grant No. 16H04788, 26116008, 20H05926, and 21K06053 to S.C., 19K16044, and 21K15020 to K.F., 23K05017 for H.T.), Institute for Fermentation, Osaka (grant G-2021-2-063 to S.C.), and by JST, ACT X (Grant No. JP1159335 to H.T.)

## Author contributions

K.F., N.T., M.Y., and S.C. designed the research, K.F., N.T., M.Y., and S.C. performed experiments, K.F. performed bioinformatic analyses, K.F., H.T., and S.C. supervised the work, all authors analyzed the data, and K.F. and S.C. wrote the manuscript.

## Competing interests

The authors declare no competing interests.
