## [Peer Review File · Nature Communications]

REVIEWER COMMENTS

Reviewer #1 (Remarks to the Author):

In this paper, Fujiwara et al study bacterial translation arrest peptides. Upon performing some bioinformatics screening on a large genome database, they identify several classes of potential peptides found across bacterial genomes and classify them according to the type of downstream gene, N-terminal motif and phylogeny. The authors then study the extent to which representative candidate sequences can effectively lead to arrest translation both in vitro and in vivo, using various assays and two different types of bacterial ribosomes and further study the possible mechanisms associated with these arrest peptides.

Overall, this is a nice and sound study that combines bioinformatics with in vitro/in vivo assays. While some of the methodology and experiments performed in vitro/in vivo fall outside my range of expertise, the rationale of the methods and results are sound and provide enough validation of the bioinformatics predictions. Overall, the findings obtained are interesting and open for some future promising direction in the field of arrest sequences. With that being said, I found a few points that deserve some clarification or improvements, as detailed below

1. In their bioinformatics screening procedure, it wasn't exactly clear to me how after the authors perform some clustering procedure with MMseq2 (1.98), this was used or not to further name/identify the different groups of arrest sequences described after, and the authors use the same terminology of cluster/group. Is there a relation between these 2? Can the authors also provide some rationale for picking the specific representative sequences?
2. Figure 5 suggests some clustering from the spacer length among same classes of arrest peptides, showing sometimes 2 if not more characteristic spacer lengths. Have the authors investigated if this was reflected on the phylogeny or some sequence motifs?
3. The choice of colours in figure 1 was confusing with some overlap between the phylum and the AP, or colors that are too close. Some more pertinent choice of colours would greatly help to interpret the figure more easily.
4. In l. 281, the authors state that they cannot provide a clear explanation for the discrepancies between results obtained in vitro/in vivo. It does not seem surprising to me to see some differences as previous studies already reported impaired ribosome processivity between different bacterial cell

free protein synthesis systems (<https://doi.org/10.1038/s41598-020-80827-8>), or translational pausing that can sometimes only occur in vivo or in vitro <https://doi.org/10.1073/pnas.1520560113>

Could the authors here refer to the existing literature and assess if the discrepancies they found is surprising or in agreement with some previous findings? Perhaps the authors could also comment on the presence or absence of EF-P, that is known to prevent stalling at polyproline doi: [10.1016/j.molcel.2017.10.014](https://doi.org/10.1016/j.molcel.2017.10.014).

5. The authors also use two different bacterial ribosomes. It would be relevant here to look at some structural differences, notably at the PTC/p-site or exit tunnel, that could explain some the different results. The authors should be able to find existing structures from the PDB or use this recent tool <https://doi.org/10.1093/nar/gkac939>

6. Regarding the results found in Translation arrest of RAPP/RAGP motif-containing proteins in *E. coli* and *B. subtilis*, have the authors tested if the sequence that derives from *B. subtilis* Ywcl induce translation arrest in vivo for *E. coli* using a similar protocol as in fig 4?

Reviewer #2 (Remarks to the Author):

The aim of this paper is a comprehensive identification of novel regulatory arrest peptides through bacterial genome mining. The authors unveil a multitude of novel Sec/YidC-related arrest peptides characterized by diverse mechanisms and a widespread phylogenetic distribution. The experiments conducted are methodologically sound, and the conclusions drawn are novel. However, despite the scientific significance of the topic, the manuscript suffers from poor organization, making it challenging for readers to follow. Furthermore, certain results require more comprehensive explanations. Specific comments are provided below:

1. In the initial section, the authors mention the identification of several uORFs containing the RAPP/RGPP motif that do not appear to be conserved among their respective bacterial orders. This observation is intriguing. Could the authors provide further details on the phylogenetic distributions, perhaps by including a figure for clarity?

2. In the "In vitro translation arrest of the RQH family members" section, the authors employ two different systems (*Ec* and *Bs* PURE systems), yielding disparate results. An explanation regarding the differences between these systems and the reason for obtaining different results is needed.

3. In the "In vivo reporter assay to determine the efficiency of the translation arrest" section, the authors note inconsistencies in the TAI values between in vitro and in vivo experiments. Could the variation between different bacterial orders account for these discrepancies? Additional insights into the observed patterns would be valuable.

4. The "Bioinformatics analysis of the length of the spacer between the protein localization signal and the arrest site" section discusses differences in the median distance between the signal and arrest site for various substrates. Are there evolutionary reasons for these disparities, and do they correlate with distinct functions? More information on this aspect would be insightful.

5. The manuscript introduces numerous self-defined terms (e.g., F150xxxxWIxxxxGIRAGP166, uC_KYxIW cluster, uDF_DGMK-stop) without adequate explanations. It is essential to provide clear descriptions for each self-defined term to ensure reader comprehension.

6. The paper does a poor job of motivating various experiments conducted as part of this study. Before describing the results of each experiment, the authors should explain what are the goals of the designed experiments.

Minor comment.

There is inconsistency in the font usage for some gene names throughout the paper (e.g., ApcA, ApdA).

Figure 2 has too many subgraphs, please only keep the essential ones and move others to supplementary materials.

General responses to the reviewers

We are grateful to the reviewers for the positive evaluation overall and helpful and constructive comments/suggestions to improve our manuscript. In response to the reviewers' comments, we modified text and figures to provide more information or to improve the presentation. We also did an additional experiment regarding the species-specificity of Ywcl to answer a question raised by reviewer 1 (comment 6), although we decided not to include the result of this experiment in our present manuscript because this paper primarily focuses on the comprehensive identification of arrest peptides encoded upstream of *sec/yidC* genes, and therefore the species-specificity of *sec/yidC*-unrelated arrest peptide is thought to be out of the scope of this study (see also point-to-point responses).

During the preparation of the revised version, we realized that the figures in the original manuscript contained data and labels that have been unintentionally shown in the figures; i.e., we did *in vitro* and *in vivo* translation of GFP-5P-LacZ, in which five consecutive prolines are sandwich-fused between GFP and LacZ (5P in Fig. 4i, and Extended Data Fig. 4g, h in the initial submission version), but during the preparation of the manuscript for the initial submission, we have decided to remove them from our first draft since they provided little additional information in the present study and also we had to comply with the length limitation of manuscript. However, although we deleted them from the main text, the data had still remained in the figures by mistake. Thus, we deleted them in the revised version.

In addition, we made some minor modifications to add more information and collected minor errors such as grammatical errors, typos, and inconsistent color usage throughout the manuscript. We also modified the layout of the figures to improve visibility.

In the revised manuscript, we provisionally cite two under-review papers (related manuscript 1 and 2, respectively). The former reports the cryo-EM structure of two previously identified arrest peptides ApdA and ApdP, which share the RAPP sequence, which is also shared by the RQH family members we identified in the present study. The latter paper reports the cryo-EM structure of SecM, which has the RAGP sequence in the arrest motif. We believe that these two currently unpublished papers provide insights into the mechanism of the ribosomal stalling and species-specificity of other RAPP-containing arrest peptides we identified in the present study and, thus, merit being cited (p. 13, line 4, p. 21, lines 2 and 8). However, we will delete the citations if the publication timing of these two under-review papers does not coincide.

Following are our point-to-point responses to the reviewers' comments (*in Italics*)

REVIEWER COMMENTS

Reviewer #1 (Remarks to the Author):

In this paper, Fujiwara et al study bacterial translation arrest peptides. Upon performing some bioinformatics screening on a large genome database, they identify several classes of potential peptides found across bacterial genomes and classify them according to the type of downstream gene, N-terminal motif and phylogeny. the authors then study the extent to which representative candidate sequences can effectively lead to arrest translation both in vitro and in vivo, using various assays and two different types of bacterial ribosomes and further study the possible mechanisms associated with these arrest peptides.

Overall, this is a nice and sound study that combines bioinformatics with in vitro/in vivo assays. While some of the methodology and experiments performed in vitro/in vivo fall outside my range of expertise, the rationale of the methods and results are sound me and provide enough validation of the bioinformatics predictions. Overall, the findings obtained are interesting and open for some future promising direction in the field of arrest sequences. With that being said, I found a few points that deserve some clarification or improvements, as detailed below

Response: We appreciate the overall positive evaluation.

1. In their bioinformatics screening procedure, it wasn't exactly clear to me how after the authors perform some clustering procedure with MMseq2 l. 98., this was used or not to further name/identify the different groups of arrest sequences described after, and the authors use the same terminology of cluster/group. Is there a relation between these 2? Can the authors also provide some rationale for picking the specific representative sequences?

Response: We identified groups of arrest peptides based on the cluster or motif search (in the case of the arrest peptides harboring the RAPP-like sequence). Thus, in the former case, a cluster became a group, whereas, in the latter case, the arrest peptides in the same order were categorized into the same group. To clarify this point, we modified the text (p. 6. lines 16-17 in Results, and p. 25. 16-17 in Methods) so the reader can understand that the group and cluster are sometimes different and that the detailed information is described in the Supplementary Methods. We selected uORFs that appeared to have typical sequences as

representatives to do subsequent individual experiments.

2. Figure 5 suggests some clustering from the spacer length among same classes of arrest peptides, showing sometimes 2 if not more characteristic spacer lengths. Have the authors investigated if this was reflected on the phylogeny or some sequence motifs?

Response: In the case of SecM, it has been reported that SecM homologs from Pasteurellales are shorter than those from Enterobacteriales (ref 42). Thus, we added the description explaining the relationship between the spacer length and the phylogeny in Discussion (p. 23, line 18 - p. 24, line 6). We also added an additional Supplementary figure (Supplementary Fig. 33), in which we show the spacer length distributions of individual RQH families of different bacterial orders, in addition to the combined data that had shown in Fig. 5d. Moreover, we added more detailed information in the legend of Fig. 5d (p. 35, lines 30-32). In response to this and a related comment from reviewer 2 (comment 4), we also added possible reasons that might have caused the bimodal spacer length distribution of SecM in Discussion (p. 23, line 18 - p. 24, line 6)

3. The choice of colours in figure 1 was confusing with some overlap between the phylum and the AP, or colors that are too close. Some more pertinent choice of colours would greatly help to interpret the figure more easily.

Response: We modified the color choice so that the phylum and the arrest peptides can be more easily recognized. We reduced the colors of phyla by highlighting only major phyla. We also used only the pale colors for phyla. We also modified the shape of the box of labels of AP and Phylum on the left and right of the phylogenetic tree to help with an intuitive understanding of the color correspondence (Fig. 1).

4. In l. 281, the authors state that they cannot provide a clear explanation for the discrepancies between results obtained *in vitro*/*in vivo*. It does not seem surprising to me to see some differences as previous studies already reported impaired ribosome processivity between different bacterial cell free protein synthesis systems (<https://doi.org/10.1038/s41598-020-80827-8>), or translational pausing that can sometimes only occur *in vivo* or *in vitro* <https://doi.org/10.1073/pnas.1520560113>. Could the authors here refer to the existing literature and assess if the discrepancies they found is surprising or in agreement with some previous findings? Perhaps the authors could also comment on the presence or absence of EF-P, that is known to prevent stalling at polyproline doi: 10.1016/j.molcel.2017.10.014.

Response: In accordance with the reviewer's suggestion, we added additional possible reasons that might explain the discrepancies between *in vivo* and *in vitro* results in Discussion. Those include the possible involvement of EF-P (p. 22, line 22 – p. 23, line 3) and an experimental issue related to the low levels of b-gal activity (p. 23, lines 7-9). We decided not to mention the possibility of the impaired ribosome processivity *in vitro* because we are currently unsure if the general lower processivity could give a good explanation for the discrepancies observed for some specific arrest peptides.

5. *The authors also use two different bacterial ribosomes. It would be relevant here to look at some structural differences, notably at the PTC/p-site or exit tunnel, that could explain some the different results. The authors should be able to find existing structures from the PDB or use this recent tool <https://doi.org/10.1093/nar/gkac939>*

Response: We added more detailed information on the difference between *E. coli* and *B. subtilis* ribosomes by citing our previous related paper showing that the structure of uL22 but not uL4 differs in the exit tunnel and that the difference in uL22 is responsible for the species-specificity of *B. subtilis* MifM (p. 21, lines 13-16). We also cited related manuscript 1, which deals with the species-specificity of ApdA and ApdP (p. 21, line 8).

6. *Regarding the results found in Translation arrest of RAPP/RAGP motif-containing proteins in E. coli and B. subtilis, have the authors tested if the sequence that derives from B. subtilis Ywcl induce translation arrest in vivo for E coli using a similar protocol as in fig 4?*

Response: Following the reviewer's suggestion, we did additional *in vivo* and *in vitro* experiments to test if the ribosome stalling of *B. subtilis* Ywcl occurs with the *E. coli* ribosome (see attached data). However, we decided not to include these results in the current manuscript, in which we would like to focus more on the comprehensive identification of the *sec/yidC*-related arrest peptides and their phylogenetic analysis. We assume that species-specificity can occur for various reasons specific to each arrest peptide. Thus, we believe that this topic is more suitable for publishing in a future paper focusing on individual arrest peptides. However, we appreciate the reviewer's suggestion that encouraged us to do this experiment, which we also wanted to know the results.

Reviewer #2 (Remarks to the Author):

The aim of this paper is a comprehensive identification of novel regulatory arrest peptides through bacterial genome mining. The authors unveil a multitude of novel Sec/YidC-related arrest peptides characterized by diverse mechanisms and a widespread phylogenetic distribution. The experiments conducted are methodologically sound, and the conclusions drawn are novel. However, despite the scientific significance of the topic, the manuscript suffers from poor organization, making it challenging for readers to follow. Furthermore, certain results require more comprehensive explanations. Specific comments are provided below:

Response: I appreciate the positive evaluation of the reviewer and helpful suggestions regarding the presentation.

1. *In the initial section, the authors mention the identification of several uORFs containing the RAPP/RGPP motif that do not appear to be conserved among their respective bacterial orders. This observation is intriguing. Could the authors provide further details on the phylogenetic distributions, perhaps by including a figure for clarity?*

Response: In response to the reviewer's suggestion, we made an additional figure focusing on the phylogenetic distributions of orphan uORF containing the RAPP-like motif (Supplementary Fig. 1c). We also added the bacterial order information in the Supplementary Fig. 1a, b, so that readers can easily see which order has which uORF.

2. *In the "In vitro translation arrest of the RQH family members" section, the authors employ two different systems (Ec and Bs PURE systems), yielding disparate results. An explanation regarding the differences between these systems and the reason for obtaining different results is needed.*

Response: We added the following sentence "Given that only the ribosome is different between Ec and Bs PURE, the difference in the arrest efficiencies should be attributed to the difference in the ribosome structure" to clarify the difference between Ec and Bs PURE systems in Results (p. 10, lines 1-3). We also added more detailed information regarding the structural difference between *E. coli* and *B. subtilis* ribosomes in Discussion (p. 21, lines 13-16: also see the Response to the comment 5 by reviewer 1).

3. *In the "In vivo reporter assay to determine the efficiency of the translation arrest" section, the authors note inconsistencies in the TAI values between in vitro and in vivo experiments. Could the variation between different bacterial orders account for these discrepancies? Additional insights into*

the observed patterns would be valuable.

Response: We added the possible involvement of EF-P and an experimental issue due to the low levels of b-gal activity as reasons that might explain the inconsistencies between in vivo and in vitro results (p. 22, line 22 – p. 23, line 3; see also the response to the comment 4 by reviewer 1), in addition to other possible explanations we have mentioned in the original version.

4. *The "Bioinformatics analysis of the length of the spacer between the protein localization signal and the arrest site" section discusses differences in the median distance between the signal and arrest site for various substrates. Are there evolutionary reasons for these disparities, and do they correlate with distinct functions? More information on this aspect would be insightful.*

Response: In response to the reviewer's suggestion, we added a whole paragraph in Discussion, where we mention possible evolutionary reasons that might have resulted in the bimodal spacer length distribution observed for SecM (p. 23, line 18 - p. 24, line 6)

5. *The manuscript introduces numerous self-defined terms (e.g., F150xxxxWlxxxxGIRAGP166, uC_KYxIW cluster, uDF_DGMK-stop) without adequate explanations. It is essential to provide clear descriptions for each self-defined term to ensure reader comprehension.*

Response: We added or modified explanations for these terms (p. 3, lines 18-19, p5, lines 5 and 10, respectively).

6. *The paper does a poor job of motivating various experiments conducted as part of this study. Before describing the results of each experiment, the authors should explain what are the goals of the designed experiments.*

Response: We added or modified the text to clarify the purpose of each experiment (p. 11, line 1 and line 14).

Minor comment.

There is inconsistency in the font usage for some gene names throughout the paper (e.g., ApcA, ApdA).

Response: We could not recognize the inconsistency in the font usage on our PC.

Figure 2 has too many subgraphs, please only keep the essential ones and move others to supplementary materials.

Response: We selected representative results for the main figure and moved the remaining to Supplementary Figure S13.

REVIEWERS' COMMENTS

Reviewer #1 (Remarks to the Author):

The authors have added some clarifications, references and results that mostly answer my comments.

I have one remaining concern about the answer from my first comment: "We selected uORFs that appeared to have typical sequences as representatives to do subsequent individual experiments. "

This sentence is vague with no additional details provided. What does typical exactly mean? What features define a typical sequence? Are there any elements in the supplementary or main text that can help understand how much variation is there in the uORFs and how one can assess what a typical sequence should be for the different families of sequences considered?

Reviewer #2 (Remarks to the Author):

The author has now addressed most of comments. However, regarding comment 6 about the goals and design of the experiment, I still find the connection between different experiments to be weak. I am unclear about the references to "p. 11, line 1, and line 14" in the manuscript, possibly due to format differences. The author frequently starts sections with "Next," but as a reader, I expect transitions like "To test the possibility of ..." It would be helpful if the author could provide either a pipeline figure to explain the experiments or a comprehensive paragraph detailing the rationale behind each experiment.

Response to reviewers' comments

Following are our point-to-point responses to the reviewers' comments (*in Italics*)

REVIEWERS' COMMENTS

Reviewer #1 (Remarks to the Author):

The authors have added some clarifications, references and results that mostly answer my comments.

I have one remaining concern about the answer from my first comment: "We selected uORFs that appeared to have typical sequences as representatives to do subsequent individual experiments. "

This sentence is vague with no additional details provided. What does typical exactly mean? What features define a typical sequence? Are there any elements in the supplementary or main text that can help understand how much variation is there in the uORFs and how one can assess what a typical sequence should be for the different families of sequences considered?

Response: No specific criteria were used to select representative uORFs, but sequences that appeared to be exceptions were not intentionally selected. We provide sequence logo and amino acid sequence alignment of each homology group in Supplementary Figures 2-12, which would help readers understand the sequence variations and assess how typical the representative uORFs we selected are.

Reviewer #2 (Remarks to the Author):

The author has now addressed most of comments. However, regarding comment 6 about the goals and design of the experiment, I still find the connection between different experiments to be weak. I am unclear about the references to "p. 11, line 1, and line 14" in the manuscript, possibly due to format differences. The author frequently starts sections with "Next," but as a reader, I expect transitions like "To test the possibility

of ..." It would be helpful if the author could provide either a pipeline figure to explain the experiments or a comprehensive paragraph detailing the rationale behind each experiment.

Response: We modified the text according to the reviewer's suggestion (lines 138-139, 194-196, 220-222, 266-267, 327-329, 360-361; shown in red).